# Galectin-9 restricts hepatitis B virus replication via p62/SQSTM1-mediated selective autophagy of viral core proteins

Kei Miyakawa[1], Mayuko Nishi[1], Michinaga Ogawa [2], Satoko Matsunaga[1], Masaya Sugiyama[3], Hironori Nishitsuji[4], Hirokazu Kimura[5], Makoto Ohnishi[2], Koichi Watashi[6,7], Kunitada Shimotohno[4], Takaji Wakita[6] & Akihide Ryo [1✉]

Autophagy has been linked to a wide range of functions, including a degradative process that defends host cells against pathogens. Although the involvement of autophagy in HBV infection has become apparent, it remains unknown whether selective autophagy plays a critical role in HBV restriction. Here, we report that a member of the galectin family, GAL9, directs the autophagic degradation of HBV HBc. BRET screening revealed that GAL9 interacts with HBc in living cells. Ectopic expression of GAL9 induces the formation of HBc-containing cytoplasmic puncta through interaction with another antiviral factor viperin, which co-localized with the autophagosome marker LC3. Mechanistically, GAL9 associates with HBc via viperin at the cytoplasmic puncta and enhanced the auto-ubiquitination of RNF13, resulting in p62 recruitment to form LC3-positive autophagosomes. Notably, both GAL9 and viperin are type I IFN-stimulated genes that act synergistically for the IFN-dependent proteolysis of HBc in HBV-infected hepatocytes. Collectively, these results reveal a previously undescribed antiviral mechanism against HBV in infected cells and a form of crosstalk between the innate immune system and selective autophagy in viral infection.

---

[1] Department of Microbiology, Yokohama City University School of Medicine, Kanagawa 236-0004, Japan. [2] Department of Bacteriology I, National Institute of Infectious Diseases, Tokyo 162-8640, Japan. [3] Genome Medical Sciences Project, National Center for Global Health and Medicine, Chiba 272-8516, Japan. [4] Research Center for Hepatitis and Immunology, National Center for Global Health and Medicine, Chiba 272-8516, Japan. [5] School of Medical Technology, Faculty of Health Sciences, Gunma Paz University, Gunma 370-0006, Japan. [6] Department of Virology II, National Institute of Infectious Diseases, Tokyo 162-8640, Japan. [7] Research Center for Drug and Vaccine Development, National Institute of Infectious Diseases, Tokyo 162-8640, Japan. ✉email: aryo@yokohama-cu.ac.jp

Hepatitis B virus (HBV) has chronically infected 350 million people worldwide, who are now at risk of developing liver cancer[1]. HBV contains a 3.2 kb partially double-stranded DNA that consists of four overlapping genes designated C (core), X, P (polymerase), and S (surface)[2]. Among the four viral proteins, the HBV core protein (HBc) plays a pivotal role in the viral life cycle, serving as the basic unit for viral capsid assembly, and is closely associated with replication of the HBV genome and biosynthesis of progeny virions[3]. HBc is also involved in multiple viral replication steps, including the formation of viral capsids and the synthetic step of viral cccDNA, which serves as a template for the transcription of viral genes. Therefore, HBc represents a potential therapeutic target against HBV infection.

The innate immune response is the first line of cellular defense against invading pathogens. The innate immune system utilizes pattern-recognition receptors to detect the invasion of pathogens and initiate host antiviral responses, consisting primarily of the production of type I and type III interferons (IFNs)[4]. The biological response to IFN is mediated by IFN receptors and results in activation of the JAK–STAT pathway, which leads to the expression of several hundred IFN-stimulated genes (ISGs). Accumulating evidence suggests that the host innate immune system, via the effects of ISGs, can suppress replication of multiple types of viruses through a wide range of mechanisms[5,6]. For HBV infection, type I IFNs trigger intracellular events that inhibit the assembly of pre-genomic RNA-containing capsids and accelerate their degradation[7]. However, it remains unknown which ISG(s) manipulate antiviral action against HBV and how this action is regulated through intrinsic cellular machinery and systems.

Xenophagy is a form of selective autophagy that targets invading pathogens, thus, operating as a cellular-level innate immune response[8]. In xenophagy, microbial components are degraded through the autophagy–lysosomal pathway. Selective autophagy is achieved through the polyubiquitination of microbial proteins or cargo components that are recognized by autophagy receptors, such as p62 (also known as SQSTM1)[9]. Autophagy receptors specifically bind to polyubiquitin chains on the target protein, and thus function as molecular adaptors for additional components of the autophagic machinery.

In host cells, incoming viruses elicit strong innate immune responses that contribute to the prevention of viral replication and its spread, suggesting a molecular link between innate immunity and xenophagy that elicits a robust mechanism for elimination of intracellular viral components. However, it remains to be determined whether host innate immunity against HBV can trigger selective autophagy for viral components.

In this study, we conducted a high-throughput bioluminescence resonance energy transfer (BRET) screen that identified galectin-9 (GAL9) as a ISG that selectively triggers autophagic degradation of HBc, thereby suppressing viral replication. Moreover, we elucidated the molecular mechanism by which GAL9 orchestrates selective autophagy against HBc, which is mediated by viperin, RNF13 and p62. Our results reveal a previously uncharacterized host innate immune response that specifically initiates selective autophagy of HBV core components to suppress viral replication.

## Results

**Identification of HBc-binding ISGs that regulate HBV production**. Based on bioinformatics analysis of the Gene Expression Omnibus database[10], we established a human cDNA library of 130 ISGs that are specifically induced by type I and type III IFNs in human hepatocytes (designated here as "hepato-ISGs"). We subsequently expressed NanoLuc-conjugated HBc and each of the hepato-ISGs in HEK293 cells and screened for protein–protein interactions by using NanoLuc-based bioluminescence resonance energy transfer (NanoBRET) in living cells (Fig. 1a). We identified proteins encoded by ten ISGs (CYP2A6, GPD2, HCP5, IFI35, IRF7, LAMP3, LGALS9, LOX, MEL, and PSMB8) as possible interactors with HBc based on their positive BRET signals (Fig. 1a). To assess the functional relevance of these host proteins in viral particle production, we transfected HepG2 cells with each of the 10 selected ISGs along with a plasmid encoding the complete HBV genome[11] (Fig. 1b). Subsequent measurement of HBV DNA in the culture supernatant by quantitative PCR revealed that ectopic expression of LGALS9 (galectin-9, hereafter GAL9) substantially decreased the release of virions containing viral DNA (Fig. 1b). As GAL9 belongs to the family of tandem repeat-type galectins, we also tested other galectin family proteins in parallel experiments. With the exception of GAL9, galectin family proteins did not exhibit HBc interaction (Fig. 1c) or viral restriction (Fig. 1d). Based on these preliminary results, we decided to further analyze GAL9 as a negative regulator of HBc.

**GAL9 promotes HBc accumulation at LC3-positive cytoplasmic puncta**. We initially asked whether GAL9 could interact with HBc from different HBV genotypes. Our immunoprecipitation assay revealed that GAL9 was associated with HBc derived from all genotypes tested (genotypes A–D) (Supplementary Fig. 1a). To determine whether GAL9 affects HBc protein stability, we co-transfected HepG2 cells with HBc and GAL9. We found that GAL9 decreased HBc protein expression, whereas HBV core mRNA expression was almost unchanged (Fig. 2a). We also observed that GAL9 reduced HBc expression in both HepG2 cells transfected with HBV genomic DNA and HepG2.2.15.7 cells with stable HBV expression[12] (Supplementary Fig. 1b).

Several galectin family proteins have been shown to regulate antimicrobial autophagy[13]. To determine whether GAL9-dependent HBc protein reduction is mediated by the autophagic pathway, we treated cells with either the lysosomal inhibitor bafilomycin A1 or the proteasome inhibitor MG132. GAL9-mediated reduction in HBc protein was mostly rescued by bafilomycin A1, but not by MG132 (Fig. 2b). Moreover, we confirmed that GAL9 was unable to degrade HBc in cells lacking ATG5, an essential molecule of autophagy (Fig. 2c). Together, these data indicate that GAL9 induces autophagic degradation of HBc.

Next, we sought to determine whether GAL9 colocalized with HBc. Consistent with previous reports[14], HBc was mainly localized in the cytoplasm in a diffuse manner and to a lesser extent in the nucleus (Fig. 2d). Notably, upon GAL9 expression, HBc localization shifted to cytoplasmic puncta, where GAL9 was also accumulated (Fig. 2d, Supplementary Figs. 1c and 2a). Next, we labeled cell membrane structures with the lipid membrane-binding dye CellMask and found that both GAL9 and HBc colocalized with CellMask (Fig. 2e, Supplementary Fig. 2b), indicating that these proteins reside on intracellular membranes. Consistent with this observation, immune-electromicroscopic analysis showed that HBc was accumulated and/or incorporated into the GAL9-positive intracellular membrane structures (Supplementary Fig. 3). Moreover, the autophagosome marker LC3 was localized at puncta containing GAL9 and HBc (Fig. 2f), and the number of HBc localizations in LC3-positive puncta was significantly increased by GAL9 expression (Supplementary Fig. 2c). Additionally, LC3 localization was observed on and around CellMask-stained membrane structures (Supplementary Fig. 2d). Further, time-lapse imaging of HBc-expressing cells revealed the dynamics of GAL9 and LC3 colocalization 8 h after gene transfection (Supplementary Fig. 2e).

Since GAL9 is an IFN-inducible protein, we next investigated whether type I IFN could trigger the accumulation of HBc in

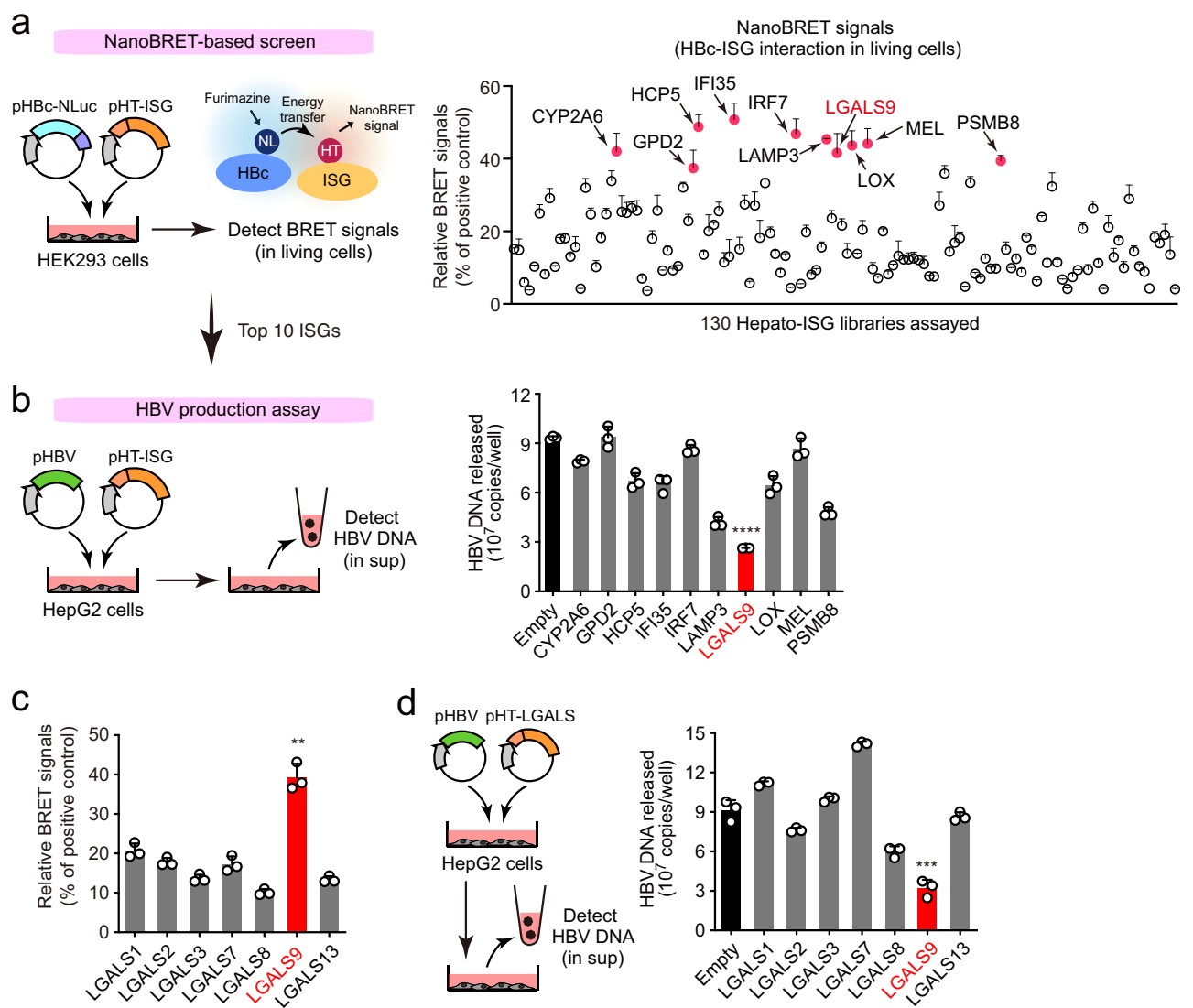

**Fig. 1 Identification of HBc-binding ISG proteins that regulates HBV production. a** NanoBRET-based screen to identify HBc-interacting proteins encoded by ISGs in living cells. HEK293 cells were co-transfected with NanoLuc (NL)-conjugated HBc (derived from HBV genotype C) and HaloTag (HT)-fused ISG expression vectors (130 genes). Following, HT-618 ligand and furimazine substrate were added. If two proteins were within 200 nm of each other, NanoBRET signals were detected. The interaction between NL-HBc and HT-HBc was used as the positive control. The top 10 proteins with the highest NanoBRET signals are indicated by arrowheads. The mean ± SEM of three independent determinations is plotted. **b** LGALS9 suppresses HBV particle production. HepG2 cells were transfected with the HBV molecular clone pHBV and either of the indicated ISG expression vectors. Cells were washed 4 h after transfection. Three days after transfection, viral DNA in culture supernatants was measured by qPCR. **c** LGALS9, but not other galectin family proteins, interacts with HBc. NanoBRET analysis as described in **a** was performed on HepG2 cells expressing NL-HBc and the indicated HT-LGALS proteins. **d** LGALS9 specifically suppresses HBV production. HBV production assay as described in **b** was performed on HepG2 cells expressing pHBV and the indicated HT-LGALS proteins. Bar charts are presented as a mean ± SD ($n = 3$ experiments). ****$P < 0.0001$ (**b**), ***$P = 0.0006$ (**d**), two-tailed unpaired $t$-test. Source data are provided as a Source Data file.

LC3-positive puncta. IFN-β increased the colocalization of HBc and LC3 at cytoplasmic foci, which was abolished by GAL9 depletion (Fig. 2g, Supplementary Fig. 2f). Consistent with this observation, the amount of HBc was reduced following IFN-β treatment in HepG2 cells expressing HBc, and this phenomenon was abolished by GAL9 depletion (Supplementary Fig. 2g). Taken together, these results suggest that type I IFN induces GAL9 expression and triggers the localization of HBc at LC3-positive autophagic puncta, leading to autophagic degradation of HBc.

**Viperin mediates HBc localization on GAL9-positive puncta.** The above results showed that HBc-puncta formation was more

prominent in IFN-β-treated cells than in GAL9-transfected cells, indicating an association with other IFN-inducible factor(s) for HBc-puncta formation via GAL9. Moreover, our in vitro pull-down analysis using recombinant proteins demonstrated that GAL9 was unlikely to bind HBc directly (Supplementary Fig. 4a), implying the existence of other molecule(s) between HBc and GAL9. We then hypothesized that other IFN-inducible factor(s) could mediate the functional interaction between GAL9 and HBc as well as regulate HBc-puncta formation. By screening the bioinformatics database (Biological General Repository for Interaction Datasets; BioGRID)[15,16], we identified five GAL9 binding proteins whose expression could be potentially induced by type I IFN (viperin, CD47, LGALS3BP, IFIT3, LAMB1).

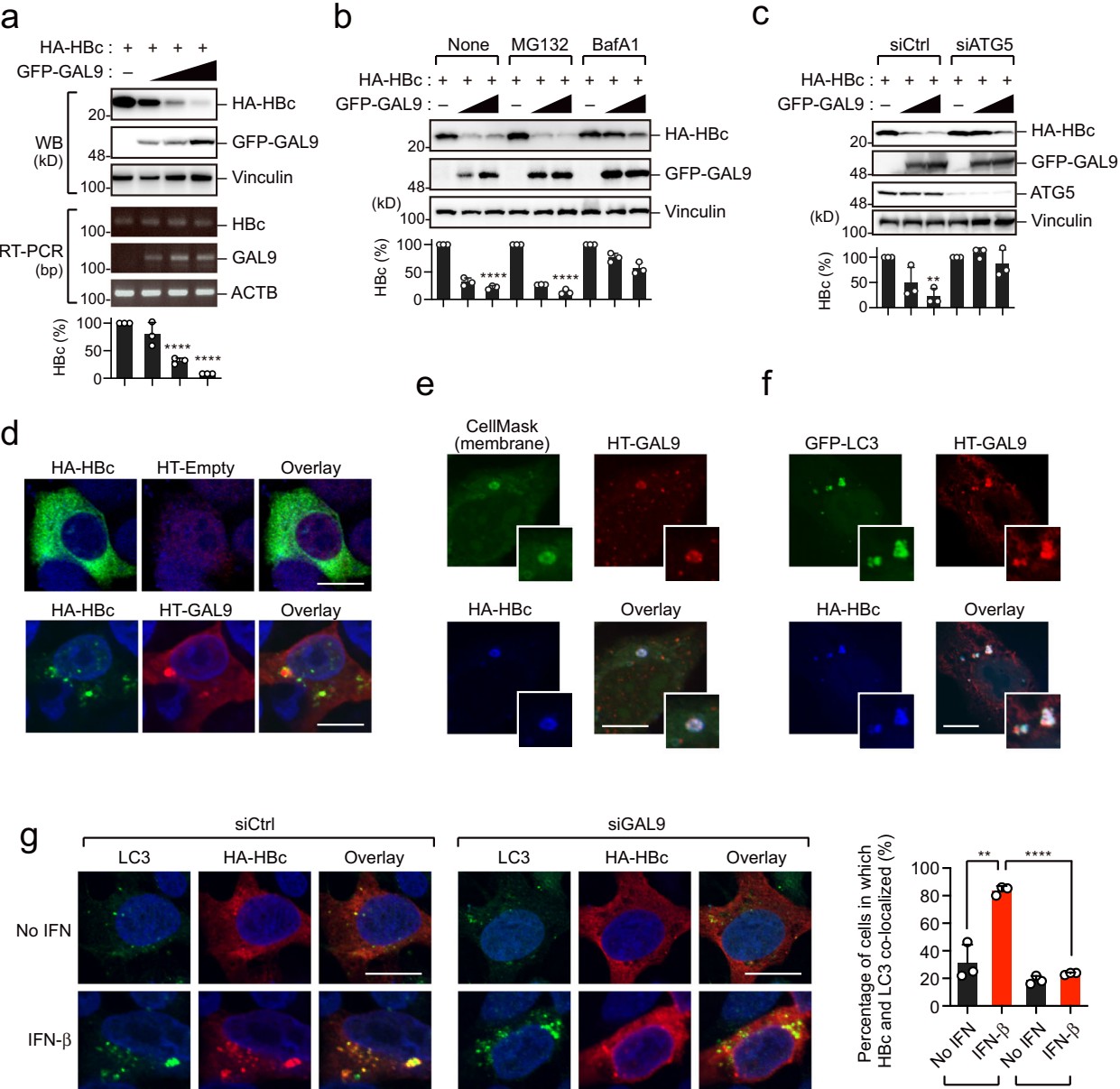

**Fig. 2 GAL9 leads to the autophagic degradation of HBc. a** GAL9 decreases the level of HBc protein. Immunoblotting (upper) and RT-PCR assays (lower) of HepG2 cells expressing HA-HBc and GFP-GAL9. **b** Bafilomycin A1 inhibits GAL9-induced HBc degradation. HepG2 cells were transfected with vectors encoding HA-HBc and GFP-GAL9. Cells were treated with bafilomycin A1 (BafA1, 100 μM) or MG132 (10 μM) 16 h prior to harvesting. **c** ATG5 depletion attenuates GAL9-induced HBc degradation. HepG2 transduced with control- (Ctrl) or ATG5-targeting siRNA were transfected with vectors encoding HA-HBc and GFP-GAL9. **d** GAL9 and HBc accumulation in cytoplasmic bodies. Confocal microscopic imaging of HepG2 cells expressing HA-HBc and HaloTag empty vector (upper) or HaloTag-GAL9 (lower). Nuclei were stained with DAPI. Scale bar, 10 μm. Another view of the cell image is shown in Supplementary Fig. 2a. **e** GAL9 and HBc accumulation in intracellular membranes. Confocal micrographs show HepG2 cells expressing HA-HBc and HaloTag-GAL9. Intracellular membranes were stained with CellMask reagent. Expanded views are also shown. Scale bar, 10 μm. Another view of the cell image is shown in Supplementary Fig. 2b. **f** GAL9 and HBc co-localize with LC3. Confocal micrographs show HepG2 cells expressing HT-GAL9, HA-HBc, and GFP-LC3. Expanded views are also shown. Scale bar, 10 μm. Another view of the cell image is shown in Supplementary Fig. 2c. **g** IFN-β increases colocalization of HBc and LC3 in a GAL9-dependent manner. Confocal micrographs show HepG2 cells expressing HA-HBc in the presence or absence of IFN-β. Scale bar, 10 μm. Another view of the cell image is shown in Supplementary Fig. 2f. The graph on the right shows the percentage of cells in which HBc localized to LC3-positive puncta ($n = 50$ cells examined over three experiments, mean ± SD). **$P = 0.0027$, ****$P < 0.0001$, two-tailed unpaired $t$-test. Scale bar, 10 μm. Bar charts in **a–c** indicate the ratio of HBc over Vinculin, as determined by densitometry, and are presented as a mean ± SD ($n = 3$ experiments). ****$P < 0.0001$ (**a**, **b**), **$P = 0.0011$ (**c**), two-tailed unpaired $t$-test. Immunoblots and micrographs are representative of experiments with similar results ($n \geq 2$). Source data are provided as a Source Data file.

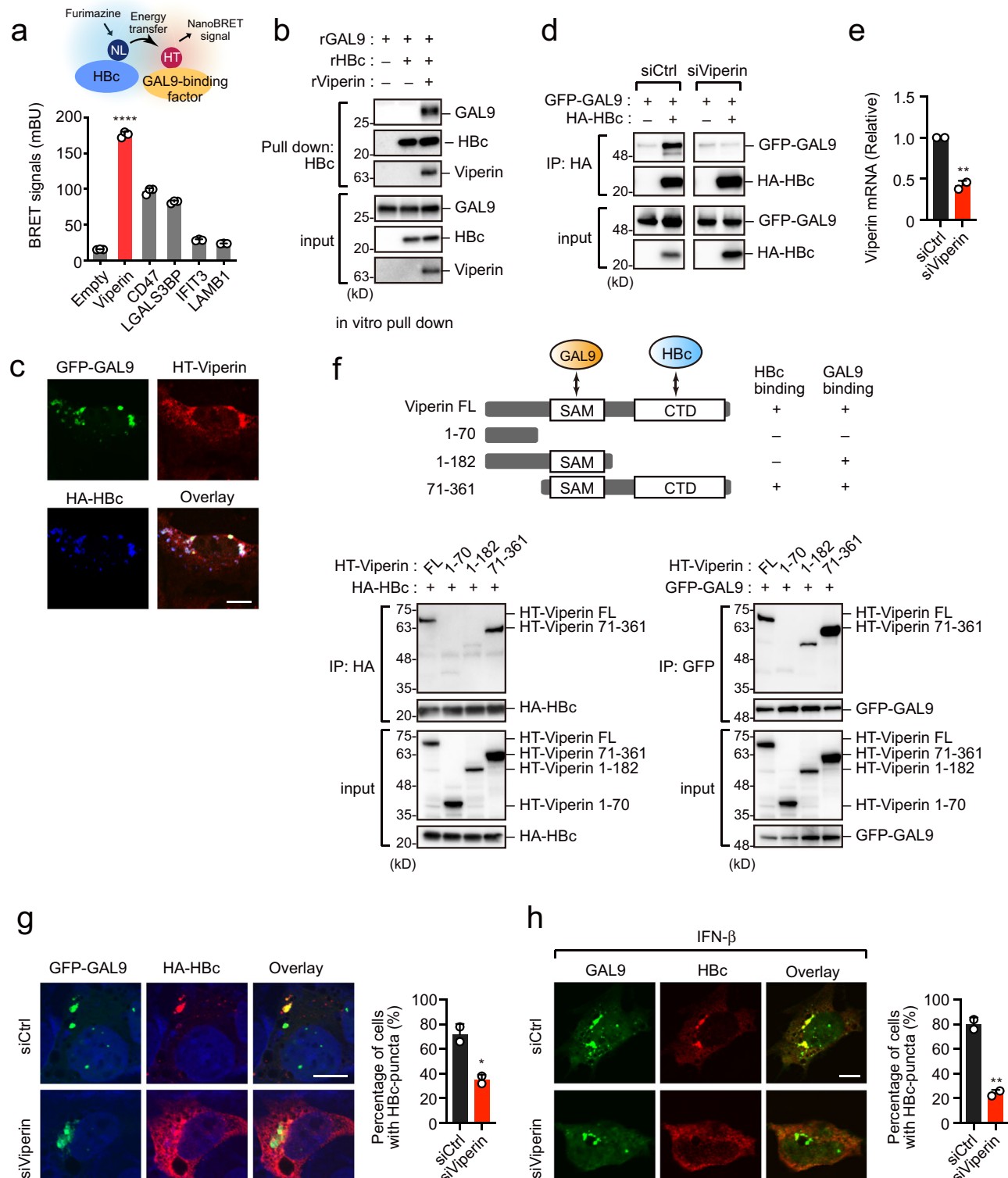

Among these, viperin was found to interact with HBc by BRET analysis (Fig. 3a). Our in vitro pull-down analysis using recombinant proteins further revealed that the interaction between HBc and GAL9 was dependent on the presence of viperin (Fig. 3b). Additionally, colocalization of viperin with HBc and GAL9 was observed in the cytoplasmic puncta (Fig. 3c, Supplementary Fig. 4b). Viperin normally localized in ER (Supplementary Fig. 4c). However, interestingly, in cells expressing HBc, the staining for ER-localized viperin was reduced (Supplementary

Fig. 4c), suggesting that HBc sequestrates viperin from ER to another cytoplasmic membrane structure where it interacts with HBc and GAL9. Immunoblotting analysis showed that viperin was expressed in hepatocytes, albeit at a low level (Supplementary Fig. 4d). Notably, knockdown of viperin significantly reduced the GAL9-HBc interaction in HepG2 cells (Fig. 3d, e). Immunoprecipitation analysis with several deletion mutants of viperin revealed that GAL9 and HBc bind to different domains, S-adenosylmethionine (SAM) domain and C-terminal domain

**Fig. 3 Viperin mediates HBc localization on GAL9-positive puncta. a** Viperin interacts with HBc and GAL9. NanoBRET analysis was performed on HepG2 cells expressing indicated HT proteins and NL-conjugated HBc ($n = 3$ experiments, mean ± SD). ****$P < 0.0001$, two-tailed unpaired $t$-test. **b** GAL9 can bind HBc in the presence of viperin. Recombinant HBc and GAL9 were incubated with or without viperin and subjected to in vitro pull-down assay using anti-HBc antibody. **c** Viperin colocalization with GAL9 and HBc. Confocal microscopic imaging of HepG2 cells expressing GFP-GAL9, HT-viperin and HA-HBc. Nuclei were stained with DAPI. Scale bar, 10 μm. Another view of the cell image is shown in Supplementary Fig. 4b. **d, e** Viperin depletion decreases the interaction between HBc and GAL9. HepG2 transduced with control- (Ctrl) or viperin-targeting siRNA were transfected with vectors encoding HA-HBc and GFP-GAL9. Cell lysates were precipitated with anti-HA antibody, followed by immunoblotting (**d**). Viperin knockdown was confirmed by RT-PCR (**e**). The mean ± SD of two independent determinations is plotted. **$P = 0.0059$, two-tailed unpaired $t$-test. **f** GAL9 and HBc bind to different domains of viperin. Immunoprecipitation assays of HepG2 cells expressing truncated viperin mutants and HA-HBc or GFP-GAL9. Cell lysates were precipitated with anti-HA or anti-GFP antibodies, followed by immunoblotting. **g** Viperin depletion attenuates HBc localization on GAL9-positive puncta. HepG2 transduced with control- (Ctrl) or viperin-targeting siRNA were transfected with vectors encoding HA-HBc and GFP-GAL9. Nuclei were stained with DAPI. The graph on the right is the percentage of HBc-puncta-forming cells ($n = 50$ cells examined over two independent experiments, mean ± SD). *$P = 0.0313$, two-tailed unpaired $t$-test. Scale bar, 10 μm. Another view of the cell image is shown in Supplementary Fig. 4e. **h** HepG2.15.7 cells transduced with control- (Ctrl) or viperin-targeting siRNA were treated with IFN-β (1000 U/mL) for 24 h. Cells were stained with anti-GAL9 and anti-HBc antibodies ($n = 50$ cells examined over two independent experiments, mean ± SD). **$P = 0.0063$, two-tailed unpaired $t$-test. Scale bar, 10 μm. Another view of the cell image is shown in Supplementary Fig. 4g. Immunoblots and micrographs are representative of experiments with similar results ($n ≥ 2$). Source data are provided as a Source Data file.

(CTD), of viperin, respectively (Fig. 3f). These data suggest that viperin plays a scaffolding role in connecting GAL9 with HBc. Consistent with this notion, GAL9-mediated HBc-puncta formation was markedly reduced in viperin-knockdown cells (Fig. 3g, Supplementary Fig. 4e). In these cells, HBc degradation by GAL9 was attenuated (Supplementary Fig. 4f). Furthermore, IFN-induced HBc-puncta formation was remarkably reduced in viperin-knockdown cells as compared with the control (Fig. 3h, Supplementary Fig. 4g, h). Taken together, these data confirm the functional significance of viperin in HBc-GAL9 puncta formation and subsequent HBc degradation.

**GAL9 directs HBc degradation via p62-dependent autophagic pathway.** The abovementioned results indicate that GAL9 initiates autophagic degradation of HBc by cooperating with viperin. Next, we investigated the molecular mechanism by which GAL9 triggers autophagy targeting HBc. Selective autophagy is a specialized type of autophagy that depends on soluble or membrane-bound autophagic cargo receptors to induce autophagosome formation on the cargo[17]. Hence, we examined the interaction of GAL9 with the autophagic cargo receptors p62 (also known as SQSTM1), NDP52, and OPTN. The results revealed that GAL9 interacted only with p62 (Fig. 4a). To test the functional role of p62 in GAL9-mediated HBc degradation, we suppressed p62 expression by using specific siRNA, and then assessed the expression of HBc and GAL9 in HepG2 cells. Targeted depletion of p62 inhibited GAL9-mediated HBc degradation (Fig. 4b).

We next investigated whether GAL9 overexpression stimulates pan-autophagy. Cells were transfected with GAL9 in the absence of HBc, and cell lysates were subjected to immunoblot analysis for endogenous p62. Our results demonstrated a slight decrease in endogenous p62 following GAL9 expression (Supplementary Fig. 5a), suggesting that GAL9 might induce pan-autophagy to a weaker degree. However, in the presence of HBc, endogenous p62 expression was markedly decreased by GAL9 expression (Supplementary Fig. 5a). These data imply that GAL9 acts on selective autophagy targeting HBc and, to a lesser extent, on pan-autophagy.

Since p62 has been shown to localize to the autophagosome formation site, we analyzed p62 localization in GAL9-expressing cells. Our results showed that GAL9 expression induced the colocalization of HBc with endogenous p62 at the cytoplasmic puncta (Fig. 4c, Supplementary Fig. 5b). Consistent with this, accumulation of HBc in p62-positive puncta was observed in IFN-treated HepG2 cells, but when endogenous GAL9 was suppressed, HBc did not show the colocalization with p62 even

after IFN treatment (Supplementary Fig. 5c). This is another evidence that IFN-induced GAL9 plays an important role in the selective autophagy targeting HBc. p62 has been reported to interact with polyubiquitinated substrates through its ubiquitin-associated domain (UBA), followed by recruitment of LC3 via the LC3-interacting region (LIR)[18]. In p62-knockdown cells, complementation with wild-type p62, but not a deletion mutant lacking UBA or LIR, could rescue GAL9-mediated HBc degradation, indicating an essential role of both domains of p62 in this process (Fig. 4d, e). Moreover, p62 has been shown to associate with other ATG proteins, including FIP200[19]. Consistently, we confirmed that FIP200 was localized at GAL9-positive puncta in HBc-expressing cells (Supplementary Fig. 5d). Together, these observations indicate that p62 plays a pivotal role in GAL9-dependent autophagic degradation of HBc.

**Role of the E3 ligase RNF13 in GAL9-mediated selective autophagy of HBc.** Considering that the ubiquitin-binding activity of p62 is indispensable for GAL9-dependent HBc degradation, we hypothesized that HBc-containing protein complexes could be polyubiquitinated during GAL9-induced autophagy. To test this possibility, we performed immunoprecipitation assays and found that the HBc immunoprecipitate was highly ubiquitinated in GAL9-expressing cells (Fig. 5a).

Next, we sought to identify the ubiquitin ligase(s) responsible for ubiquitination of the HBc complex and required for the recognition of p62. Database scanning using BioGRID revealed that GAL9 binds to the ubiquitin ligase RNF13 (Supplementary Fig. 6a). Subsequent immunoprecipitation analysis revealed that RNF13 specifically interacted with GAL9 (Fig. 5b). We also confirmed the colocalization of GAL9 and RNF13 in HepG2 cells (Fig. 5c, Supplementary Fig. 6b). This colocalization was partially observed on the cytoplasmic membranes (Supplementary Fig. 6c). Moreover, we found that RNF13 was highly glycosylated in hepatocytes (Supplementary Fig. 6d).

Galectins are carbohydrate-binding proteins with regulatory functions in intracellular glycoprotein traffic via their carbohydrate recognition domain (CRD)[13]. We then investigated whether GAL9 binds to glycosylated RNF13. Cell lysates were treated with or without the deglycosylation enzyme mix, and then subjected to immunoprecipitation analysis. Our results showed that RNF13 glycosylation was a prerequisite for GAL9 interaction (Fig. 5d). Moreover, immunoprecipitation analysis with the deletion mutant lacking CRD revealed that GAL9 could bind RNF13 through CRD (Fig. 5e). GAL9 has two CRDs on each terminus, both of which were able to bind RNF13 (Supplementary Fig. 6e).

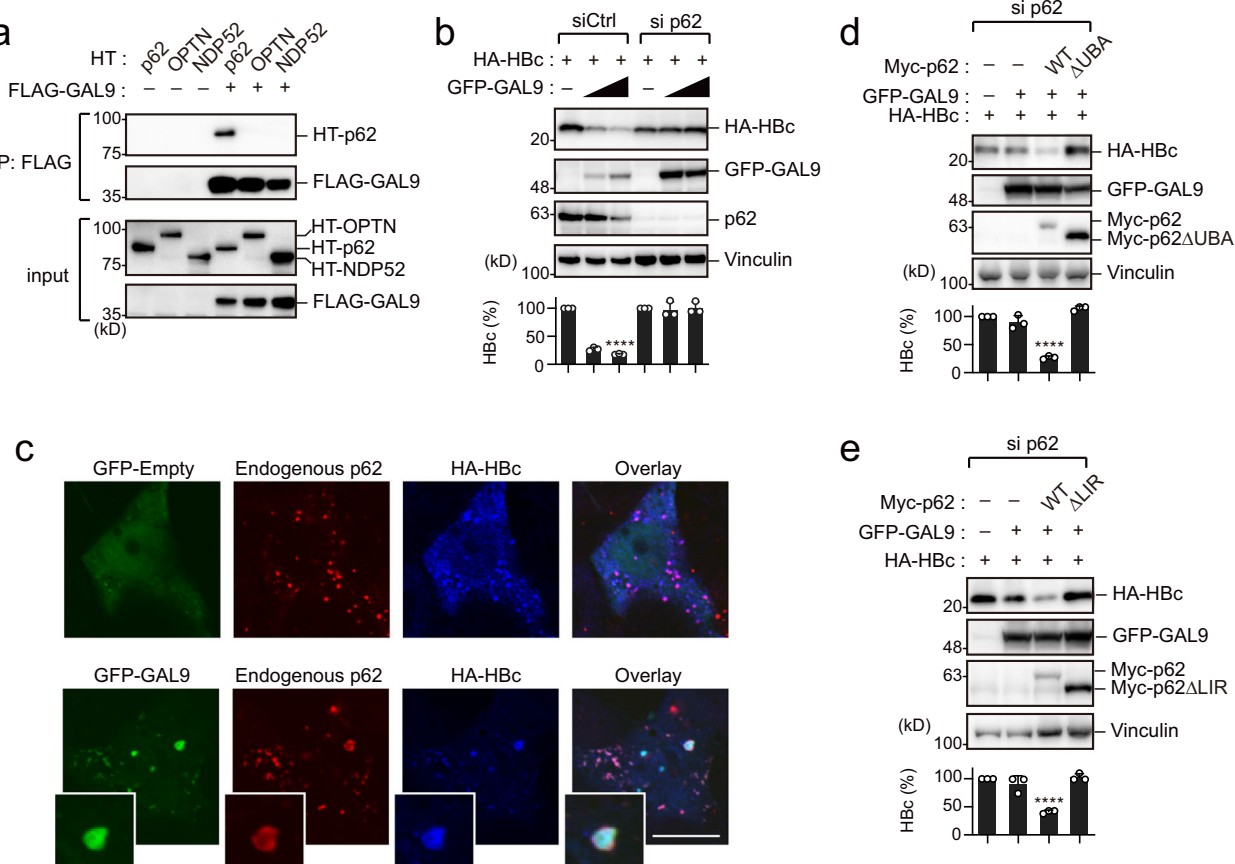

**Fig. 4 GAL9 degrades HBc via p62-mediated selective autophagy. a** GAL9 interacts with p62. Immunoprecipitation assay was performed on HepG2 cells expressing FLAG-GAL9 and indicated HT-fused proteins. Cell lysates were precipitated with anti-FLAG antibody, followed by immunoblotting. **b** p62 is required for GAL9 activity. HepG2 cells were transduced with control- (Ctrl) or p62-targeting siRNA, and then transfected with vectors expressing HA-HBc and GFP-GAL9. Cells were subjected to immunoblotting analysis to detect the indicated proteins. **c** GAL9 promotes colocalization of p62 and HBc. Micrographs show HepG2 cells expressing FLAG-HBc (blue) with or without GFP-GAL9 (green). Endogenous p62 was stained in red. Scale bar, 10 μm. Another view of the cell image is shown in Supplementary Fig. 5b. **d**, **e** The ubiquitin-binding domain (UBA) and LC3-interacting region (LIR) of p62 are required for GAL9-mediated HBc degradation. HepG2 cells were transduced with p62-targeting siRNA, and then transfected with vectors expressing HA-HBc, GFP-GAL9, and siRNA-resistant Myc-p62 (WT, ΔUBA, or ΔLIR). Cells were subjected to immunoblotting analysis to detect the indicated proteins. Bar charts in **b**, **d**, **e** indicate the ratio of HBc over Vinculin, as determined by densitometry, and are presented as a mean ± SD ($n = 3$ experiments). ****$P < 0.0001$, two-tailed unpaired $t$-test. Immunoblots and micrographs are representative of experiments with similar results ($n \geq 2$). Source data are provided as a Source Data file.

In addition, RNF13 apparently did not affect the oligomerization of GAL9 (Supplementary Fig. 6f).

Our next functional analysis demonstrated that targeted depletion of RNF13 significantly attenuated GAL9 activity in terms of HBc degradation (Supplementary Fig. 6g, h), as well as polyubiquitination of the HBc complex but not HBc itself (Fig. 5f). Our in vitro ubiquitination analysis showed that RNF13 was polyubiquitinated as a mode of auto-ubiquitination, and this process was markedly increased by GAL9 (Fig. 5g). Notably, polyubiquitination of the RNF13 mutant W270A, devoid of E3 ligase activity[20,21], was not induced by GAL9 (Supplementary Fig. 6i), again confirming that the auto-ubiquitination of RNF13 was triggered by GAL9. Consistently, re-expression of wild-type RNF13, but not the W270A mutant, rescued GAL9-induced HBc degradation in RNF13-depleted cells (Supplementary Fig. 6j). Notably, we found that expression of wild-type GAL9, but not CRD-deleted mutant unable to bind RNF13, could enhance both RNF13 auto-ubiquitination and p62 association (Fig. 5h). Together, these data indicate that GAL9 enhances auto-ubiquitination of RNF13, which subsequently leads to the recruitment of p62 and eventually gives rise to the selective autophagy targeting HBc.

**GAL9 acts as an IFN-inducible antiviral factor**. To determine whether GAL9 affects HBV replication, we generated HepG2 cells with tetracycline-inducible expression of GAL9 (HepG2-tet-GAL9 cells). We transfected HepG2-tet-GAL9 cells with a plasmid encoding the complete genome sequence of HBV, and then treated them with the tetracycline analog doxycycline (dox) to induce GAL9 expression. Notably, dox treatment decreased the levels of intracellular HBc in a dose-dependent manner, but not of the viral surface protein HBs (Fig. 6a, Supplementary Fig. 7a). Consistent with this, dox-induced GAL9 decreased the amounts of HBc antigen and viral DNA in the cell supernatant (Fig. 6b). These results suggest that GAL9 inhibits the production of infectious viral particles by decreasing the amount of cytosolic HBc.

To assess the impact of GAL9 on HBV replication, we transduced GAL9 into primary human hepatocytes (PHHs) by using a lentiviral vector and infected the transfected cells with HBV (Fig. 6c). GAL9 expression significantly decreased the release of infectious virions (Fig. 6d, e) and intracellular cccDNA levels (Fig. 6f), presumably due to reduced capsid recycling. Interestingly, analysis of recombinant HBV expressing the NanoLuc reporter[22] revealed that GAL9 had only a modest

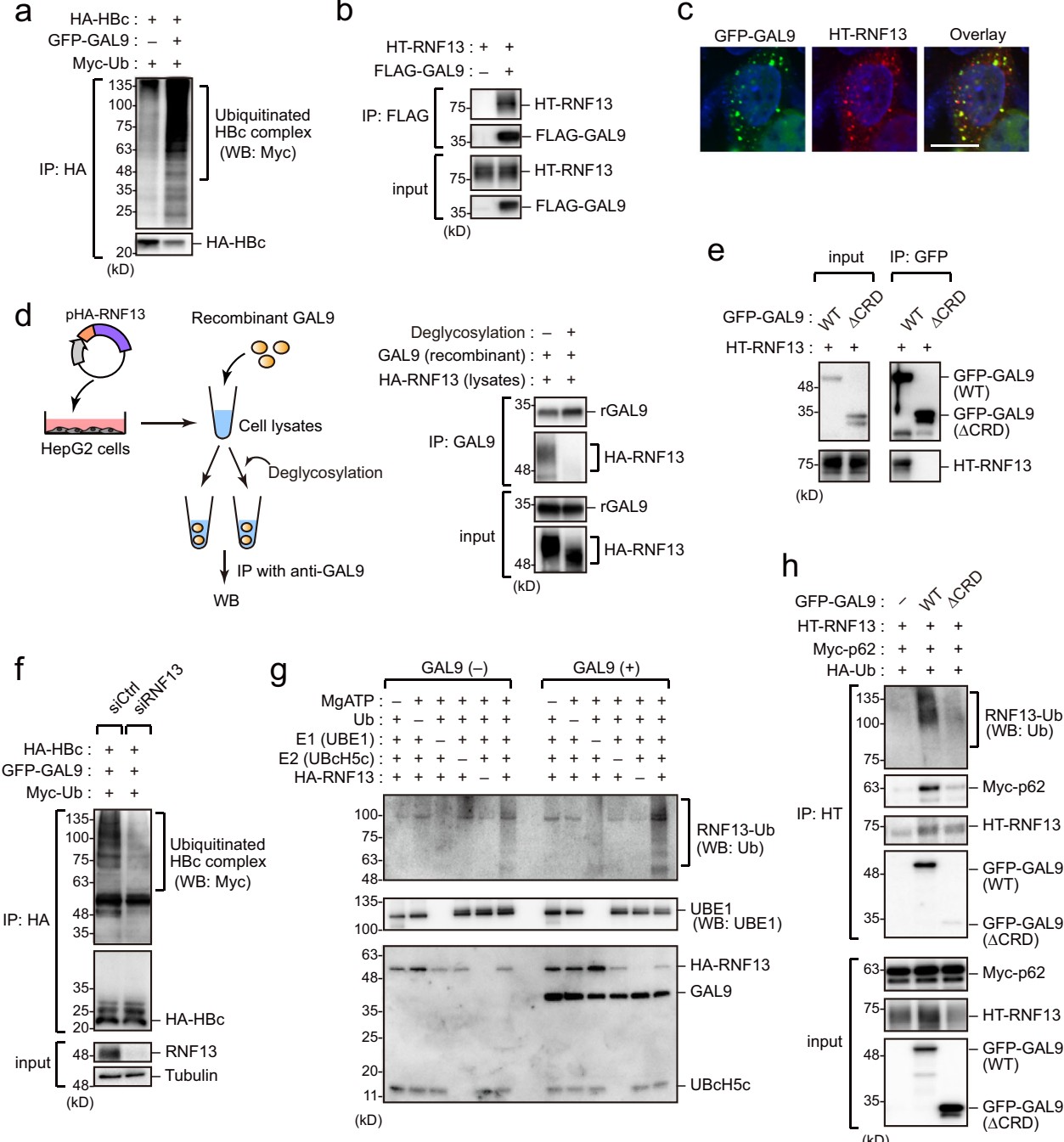

**Fig. 5 RNF13 regulates GAL9/p62-mediated selective autophagy targeting HBc. a** GAL9 induces HBc-complex ubiquitination. HepG2 cells expressing HA-HBc, GFP-GAL9, and Myc-Ubiquitin (Ub) were pulled down with anti-HA antibody, and the pulldowns were analyzed by immunoblotting. **b** GAL9 interacts with RNF13. Immunoprecipitation assay was performed on HepG2 cells expressing FLAG-GAL9 and HT-RNF13. Cell lysates were precipitated with anti-FLAG antibody, followed by immunoblotting. **c** Confocal microscopic imaging of HepG2 cells expressing GFP-GAL9 and HT-RNF13. Nuclei were stained with DAPI. Scale bar, 10 µm. Another view of the cell image is shown in Supplementary Fig. 6b. **d** Glycosylation of RNF13 is required for GAL9 interaction. HepG2 cells expressing HA-RNF13 were immunoprecipitated with anti-HA. The immunoprecipitates were incubated with recombinant GAL9 in the presence or absence of Protein Deglycosylation Mix II (5 units/µL) and then pulled down with GAL9 antibody for immunoblotting. **e** GAL9 CRD binds to RNF13. Immunoprecipitation assays of HepG2 cells expressing GAL9 CRD-deleted mutant and HT-RNF13. Cell lysates were precipitated with anti-GFP antibody, followed by immunoblotting. **f** RNF13 is a cofactor for ubiquitination of HBc complex. HepG2 cells expressing HA-HBc, GFP-GAL9, and Myc-Ub. Cells were transduced with siRNA targeting RNF13 at 24 h prior to DNA transfection. Cells were then immunoprecipitated with anti-HA antibody and subjected to immunoblotting. **g** Ubiquitination of RNF13 is increased by GAL9. HepG2 cells expressing HA-RNF13 were immunoprecipitated with anti-HA. The immunoprecipitates were incubated with indicated recombinant proteins in the presence or absence of GAL9 for 1 h. Samples were then electrophoresed and the gel was subsequently processed for CBB staining or immunoblotting with anti-Ub and UBE1 antibodies. **h** CRD-deleted GAL9 does not enhance the auto-ubiquitination of RNF13 and p62 recruitment. HepG2 cells expressing indicated proteins were precipitated with anti-HT antibody, followed by immunoblotting. Immunoblots and micrographs are representative of experiments with similar results ($n \geq 2$). Source data are provided as a Source Data file.

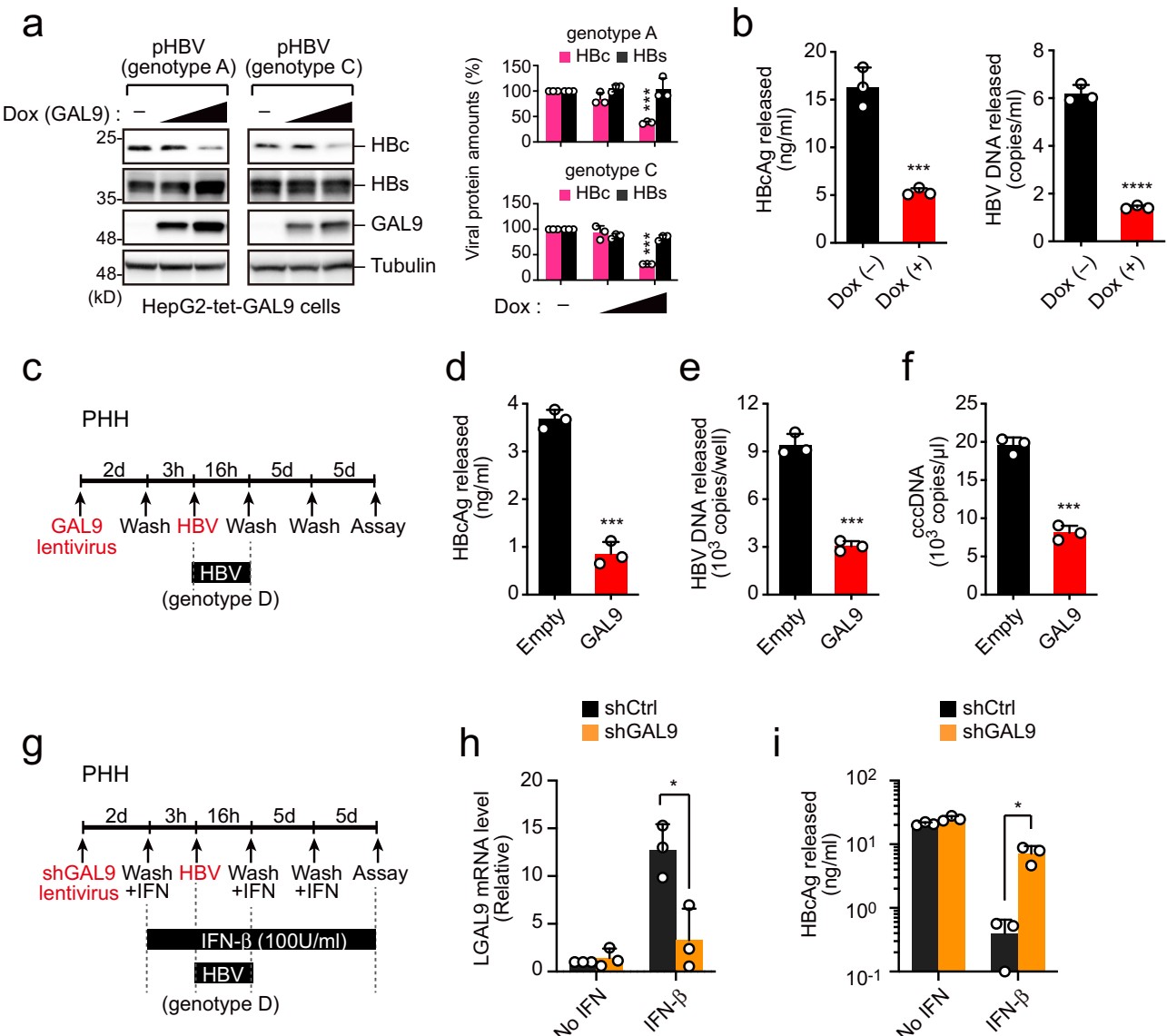

**Fig. 6 GAL9 acts as an IFN-inducible antiviral factor. a** Induced GAL9 degrades intracellular HBc. HepG2-tet-GAL9 cells were transfected with pHBV genotype A or C, and then were treated with doxycycline (0.5 or 1 µg/mL) to induce GAL9. Cells were subjected to immunoblotting analysis to detect the indicated proteins. Bar charts indicate the ratio of HBc or HBs over Vinculin, as determined by densitometry, are presented as a mean ± SD ($n = 3$ experiments). ****$P < 0.0001$, two-tailed unpaired $t$-test. **b** Induced GAL9 inhibits HBV particle release. HepG2-tet-GAL9 cells were transfected with pHBV (genotype C). Four hours after transfection, the cells were washed three times and treated with doxycycline. Three days after transfection, supernatants were harvested, followed by the detection of HBcAg and viral DNA. Bar charts are presented as a mean ± SD ($n = 3$ experiments). ***$P = 0.0008$, ****$P < 0.0001$, two-tailed unpaired $t$-test. **c–f** GAL9 inhibits HBV replication. The experimental design is presented in **c**. Primary human hepatocytes (PHHs) were transduced with lentiviral vectors carrying a GFP empty vector or GFP-GAL9 2 days before HBV infection. Eleven days after infection, supernatants were subjected to HBcAg ELISA (**d**) and qPCR (**e**) to detect released infectious HBV. The cells were subjected to qPCR to detect the cccDNA (**f**). Bar charts are presented as a mean ± SD ($n = 3$ experiments). ***$P = 0.0001$ (**d**, **e**), ***$P = 0.0002$ (**f**), two-tailed unpaired $t$-test. **g–i** GAL9 depletion attenuates IFN antiviral activity. The experimental design is shown in (**g**). PHHs transduced with control (Ctrl) or GAL9-targeting shRNA were infected with HBV in the presence or absence of IFN-β (100 U/mL). Eleven days after infection, the cells were subjected to qPCR to detect GAL9 expression (**h**). The supernatants were subjected to HBcAg ELISA to detect the released HBV (**i**). Bar charts are presented as a mean ± SD ($n = 3$ experiments). *$P = 0.0192$ (**h**), *$P = 0.035$ (**i**), two-tailed unpaired $t$-test. Immunoblots are representative of experiments with similar results ($n \geq 2$). Source data are provided as a Source Data file.

inhibitory effect on the early phase of virus infection (Supplementary Fig. 7b). These results suggest that GAL9 can inhibit the late phase of the viral life cycle, including viral production and capsid recycling processes, more efficiently than the early phase of viral infection.

Finally, we investigated the effect of type I IFN (IFN-β) on GAL9 expression and HBV core formation. PHHs were

transduced with shRNA targeting GAL9 and then infected with HBV (Fig. 6g). IFN-β significantly upregulated GAL9 expression by >10 fold (Fig. 6h). In addition, IFN-β prominently decreased viral core release, but this reduction was attenuated by GAL9 knockdown (Fig. 6i). This trend was also observed in the supernatant HBV DNA and intracellular cccDNA (Supplementary Fig. 7c). We performed parallel analyses with PHHs stably

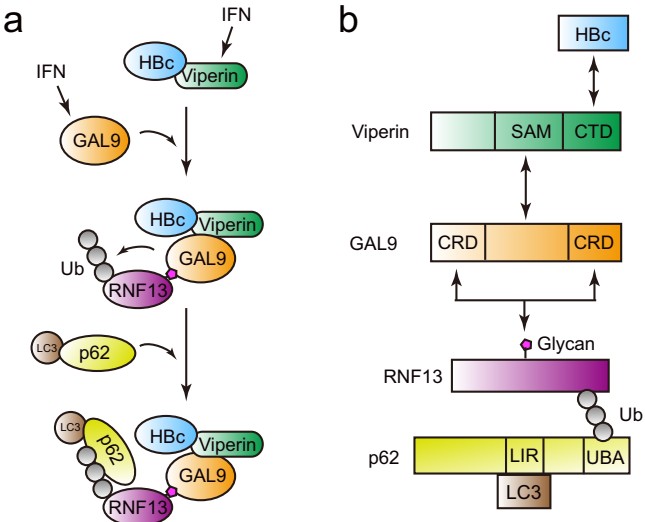

**Fig. 7 Model of GAL9-induced HBc degradation. a** Molecular mechanism of selective autophagy of HBc by GAL9. Type I IFN induces expression of GAL9 and viperin, which bind to RNF13 and HBc, respectively. Viperin binds to HBc and GAL9, leading to the HBc-puncta formation. Auto-ubiquitination of RNF13 is enhanced by GAL9, and autophagosome formation occurs when p62 is recruited. **b** Detailed binding mechanism. Viperin binds to GAL9 and HBc in the SAM and C-terminal domains, respectively. The CRD domain of GAL9 is important for binding to RNF13 carbohydrate chains. Ubiquitination of RNF13 is recognized by the UBA domain of p62.

expressing shRNA targeting RNF13, p62, or ATG5 and confirmed that knockdown of these molecules also attenuated the antiviral effects of IFN-β (Supplementary Fig. 7d). Taken together, these results indicate that IFN-induced GAL9 and its auxiliary factors, such as RNF13 and p62, cooperatively act to reduce HBV production in primary hepatocytes.

## Discussion

In this study, we identified a molecular mechanism underlying the antiviral immune system that is intimately linked with autophagic degradation of HBV core protein. Our results reveal that GAL9 is an IFN-inducible antiviral protein that triggers HBc-puncta formation by interacting with another antiviral protein viperin. Moreover, we showed that GAL9 facilitates the auto-ubiquitination of RNF13, resulting in the recruitment of p62 and LC3 to initiate autophagy. (Fig. 7a, b). Our findings thus provide a previously undescribed molecular function of GAL9 as an innate antiviral factor and shed a light on the development of therapeutic strategies to reduce progeny virion production in HBV-infected cells.

Galectin family proteins are involved in multiple immunological processes, such as immune cell activation, cytokine production, inflammation, and cancer immunity[23,24]. Galectins are primarily localized in the cytoplasm and are also secreted in the extracellular space to play key roles in the modulation of immune system[23]. GAL9 was originally identified as a chemoattractant that recruits eosinophils into tissues during inflammatory reactions[25]. Extracellular GAL9 binds Tim-3, a Th1 cell-specific surface protein, to play an anti-inflammatory role by inducing cellular apoptosis. Indeed, GAL9 suppresses Th1 autoimmunity by inducing IFN-γ-mediated cell death in activated Th1 and Th17 cells[26]. In addition, GAL9 levels in serum from patients infected with HBV are significantly elevated[27,28], implying that the protein plays a role in modulating antiviral immunity. However, the physiological role of intracellular GAL9 in HBV infection remains unclear. Our comprehensive protein–protein interaction analysis and subsequent

functional validations revealed that intracellular GAL9 associates with HBc and viperin at cytoplasmic foci and induces autophagic clearance of this viral protein, thereby decreasing HBV propagation. Moreover, we found that the antiviral effects of endogenous GAL9 were evident in hepatocytes treated with type I IFN. Of note, the effect of type I IFN on HBV propagation was significantly decreased by GAL9 depletion, illustrating the potential antiviral activity of GAL9. Together, these results suggest that intracellular GAL9 is an IFN-inducible innate antiviral factor that elicits the host autophagic pathway to selectively degrade the viral core protein in cooperation with viperin.

Human IFNs are known to prevent HBV replication in vitro and in vivo[7,29–31]. Indeed, IFNs induce the expression of an array of ISGs with anti-HBV activity. For example, indoleamine 2,3-dioxygenase (IDO) is an ISG that can decrease the level of intracellular HBV DNA[6]. In addition, the IFN-inducible protein tetherin/BST2 inhibits HBV secretion[32–34]. In regard to ISGs targeting HBc, Li et al. demonstrated that Myxovirus Resistance Gene A (MxA) interacts with HBc, and that the complex accumulates at the perinuclear compartment, thereby suppressing formation of infectious HBV particles[35]. In this study, we revealed that GAL9 is a potent ISG that restricts HBV replication. We also found that another interferon stimulatory protein, viperin, plays an important role in GAL9 targeting to HBc. Although we could not detect the stable interaction of viperin with HBc in the initial NanoBRET screening, probably due to specific experimental conditions, our immunoprecipitation and immunofluorescent analyses revealed that viperin bound both HBc and GAL9 and colocalized at the GAL9/HBc-positive cytoplasmic puncta. Viperin is a potent antiviral protein that limits the replication of both DNA and RNA viruses[36]. Viperin has an amphiphilic α-helix domain at the N-terminal side and is known to be present in the endoplasmic reticulum and lipid droplets[37]. Moreover, viperin plays a pivotal role in the production of type I IFN from plasmacytoid dendritic cells, in addition to its direct inhibition of viral replication[38]. Our current study further revealed a distinct function of viperin in the autophagic degradation of HBc through its interaction with GAL9. Since recent studies have introduced the cooperative interaction between IFN-inducible factors in antiviral processes[39], the functional association between GAL9 and viperin could be a new mode of antiviral immunity targeting viral proteins.

In targeted autophagy, ubiquitinated proteins are often recognized by selective autophagy receptors of the p62/SQSTM1-like receptor (SLR) family[40]. SLR family proteins contain multiple domains, including a ubiquitin-associated domain and an LC3-interacting region, that serve as scaffolding functions in the initiation of selective autophagy of ubiquitin-conjugated substrates[40]. The literature indicates that SLR and galectin family proteins interact functionally. For example, galectin-8 (GAL8), which has the same tandem repeat-type structure as GAL9, recruits the autophagy receptor NDP52 to glycans on damaged membranes of *Salmonella*-containing vesicles in a ubiquitin-dependent manner, which triggers autophagic activation, resulting in the confinement of bacteria within autophagosomes[41]. Our findings demonstrate that GAL9 selectively associates with glycosylated RNF13, thereby promoting the auto-ubiquitination of RNF13 during its association with the HBc-viperin complex. The interaction between GAL9 and RNF13 may occur on the cytoplasmic membranes, but it remains elusive whether the interaction occurs on the lumenal or the cytoplasmic side. Because GAL8 could associate with the glycans exposed on the lumenal side upon membrane damage[41,42], GAL9 may associate with glycosylated RNF13 via ruptured membrane. Although the molecular switch for RNF13 glycosylation is still elusive, our findings indicate that a series of protein–protein interaction and post-translational modifications could trigger the autophagic degradation of HBc.

In conclusion, this study revealed that GAL9 and its auxiliary factors play an important role in intracellular antiviral immunity, mediated by host-selective autophagy targeting the viral core protein during HBV infection. Strategies for augmenting antiviral activity induced by GAL9 may provide a therapeutic interventions against HBV. Moreover, a better understanding of the molecular mechanisms by which HBV evades the IFN-induced host immune response during a relatively long period of chronic infection may help to develop more effective strategies for combating HBV in combination with existing antiviral therapies.

## Methods

**Plasmids**. In this study, pUC19-C_JPNAT (genotype C) and pUC19-Ae_US (genotype A)[11] were used as the HBV molecular clones. HBc cDNA was amplified from pUC19-C_JPNAT with the appropriate primer pairs, followed by subcloning into pcDNA-based vector and pNLF1-C (Promega). Expression vectors encoding N-terminal HaloTag-conjugated proteins were prepared by Kazusa Genome Technologies and purchased from Promega. Human GAL9 cDNA was amplified from the HaloTag-GAL9 expression vector (Kazusa Genome Technologies) and subcloned into the pcDNA-based vector and pEGFP or pmCherry vector (Takara Bio). GAL9, viperin, and RNF13 mutants were constructed using PCR-based mutagenesis. The expression plasmids for GFP fused to LC3 were obtained from Addgene (#11546). Myc-tagged ubiquitin and p62 expression plasmids have been described previously[43,44].

**Cells and RNA interference**. HEK293 cells (ATCC, #CRL-1573) were cultured in DMEM (Wako, #043-30085) supplemented with 10% fetal bovine serum (FBS). HepG2 cells (JCRB, #JCRB1054) and HepG2.2.15.7 cells[12] were maintained on collagen-coated dishes with DMEM/F-12 GlutaMAX (Thermo, #10565018) supplemented with 10% FBS, 10 mM HEPES, and 5 μg/mL insulin. To generate HepG2-tet-GAL9 cells, HepG2-Tet-On Advanced cells (Takara Bio, #630932) were transduced with a retroviral vector encoding the GAL9 gene fused to a tetracycline-responsive element, and then selected with 1 μg/mL puromycin. PHHs were purchased from PhoenixBio (#PPC-P12). For transient knockdown, HepG2 cells were transfected with mixed gene-specific siRNAs (Qiagen, #SI03188409 and #SI04259066 for GAL9, #SI00057596 and #SI03089023 for p62, #SI00096229 and SI00096236 for RNF13, #SI00069265 and #SI02633946 for ATG5, #SI00708484 and #SI04354700 for viperin) or control siRNA (Qiagen, #1027281) by using Lipofectamine RNAiMAX Transfection Reagent (Thermo, #13778030). Alternatively, PHHs were transduced with lentiviral particles carrying gene-specific shRNA (Santa Cruz, #sc-35444-V for GAL9, #sc-78079-V for RNF13, #sc-29679-V for p62, #sc-41445-V for ATG5).

**NanoBRET-based protein–protein interaction assay**. HEK293 cells in white 96-well plates were transfected with vectors encoding HaloTag-fused protein (100 ng) and NanoLuc-fused protein (1 ng)[45]. Following, HT-618 ligand and substrate were added at 36 h and 48 h after transfection. NanoBRET activity was measured using the NanoBRET Nano-Glo Detection System (Promega, #N1661) on a GloMax Discovery system (Promega, software version 3.2.3).

**Transfection-based HBV production assay**. HepG2 cells in six-well plates were co-transfected with pUC19-C_JPNAT (1 μg) and ISG expression vectors (1.5 μg) by using Lipofectamine 3000 (Thermo, #L3000001). Alternatively, HepG2-tet-GAL9 cells in six-well plates were transfected with HBV molecular clone (1.5 μg). Four hours after transfection, the cells were washed three times with fresh medium to remove extracellular DNA. In the experiments with HepG2-tet-GAL9, cells were cultured with medium including 0.5 or 1 μg/mL of doxycycline. Three days post-transfection, cell debris was cleared from the culture supernatants by centrifugation at 860 × g for 3 min. Viral DNA was extracted using the QIAamp DNA Blood Mini kit (Qiagen, #51104) and then quantified by real-time PCR using TB Green Premix Ex Taq II (Takara Bio, #RR820S) on a CFX-96 system instrument (Bio-Rad). The primer pairs used in this study are listed in Supplementary Table 1. The levels of HBcAg in each supernatant were measured using the QuickTiter HBcAg ELISA kit (Cell Biolabs #VPK-150).

**HBV infection assay**. HBV was derived from the supernatants of HepG2.2.15.7 cells[12], which stably expressed the complete HBV genome (genotype D). The collected supernatants were filtered through a 0.45 μm filter (Merck, #SLHV033RB) and concentrated using the PEG Virus Precipitation Kit (BioVision, #K904-50). PHHs in 24-well plates were infected with HBV (500 GEq/cell). Ten days after infection, the culture supernatants were harvested and subjected to quantification of viral DNA and HBcAg, as described above. In experiments using IFN, IFN-β (100 U/mL; Wako, #092-06061) was added to cultures 3 h before infection. Total RNA and viral DNA were extracted with RNeasy mini kit (Qiagen, #74104) and QIAamp DNA Blood Mini kit (Qiagen, #51104), respectively[12,46]. Gene

quantification was done by real-time PCR as described above. The primer pairs used in this study are listed in Supplementary Table 1.

**HBc degradation analysis**. HepG2 cells in 12-well plates were co-transfected with pcDNA-HA-HBc (200 ng) and pEGFP-GAL9 (400–800 ng), with or without the p62 or RNF13 expression vector (400 ng). In siRNA-based gene knockdown experiments, cells were transfected with gene-specific siRNAs (20 pmol) one day before DNA transfection. In experiments using compounds, cells were treated with bafilomycin A1 (100 μM; Merck #196000) or MG132 (10 μM; Merck, #474790) 16 h prior to harvest. Two days after HBc transfection, cells were harvested with SDS sample buffer, loaded onto 10–20% gradient gels (Wako, #198-15041), and blotted onto PVDF membranes (Merck, #IPVH00010). Membranes were probed with primary antibodies and horseradish peroxidase–conjugated secondary antibodies. The antibodies used in this study are listed in Supplementary Table 2. Proteins were visualized on a FluorChem digital imaging system (Alpha Innotech) or a LuminoGraph imaging system (ATTO, ImageSaver software version 6.0), and band intensities were quantified using ImageJ software version 1.4 (NIH). Alternatively, to detect HBc RNA, cells were subjected to mRNA extraction and subsequent cDNA synthesis using the RNeasy mini kit (Qiagen, #74104) and ReverTra Ace (Toyobo, TRT-101), respectively. The primer pairs used in this study are listed in Supplementary Table 1.

**Immunoprecipitation and ubiquitination analysis**. For in immunoprecipitation assays, HepG2 cells in six-well plates were transfected with vectors encoding tagged proteins[45]. At 48 h post-transfection, cells were lysed with HBST buffer (10 mM HEPES pH 7.4, 150 mM NaCl, 0.5% Triton-X-100) containing protease inhibitor (Merck, #4693124001). Cell lysates were immunoprecipitated with 2 μg of antibodies and 10 μl of protein G Sepharose (Cytiva, #17061801) for 16 h. Alternatively, EZview Red Affinity Gel (Merck, #E6779, F2426, E6654), GFP-Trap Agarose (Chromotek, #gta-10), or Halo-Trap (Chromotek, #ota-10) were used. Bound proteins were washed with HBST buffer and analyzed by immunoblotting as described above. For in cell ubiquitination assays, HepG2 cells in six-well plates were transfected with vectors encoding HA-HBc (500 ng), GFP-GAL9 (1 μg), and Myc-ubiquitin (1 μg) in the presence or absence of RNF13 expression plasmids (500 ng). Cells were treated with 100 μM bafilomycin A1 and 2 μM MG132 for 16 h before harvest. Cell lysates were immunoprecipitated, and bound proteins were analyzed by immunoblotting as described above. For in vitro pull-down assays, each recombinant protein was incubated at 37 °C for 1 h in 200 μL Tris buffer (pH 7.5) and immunoprecipitated with antibodies (2 μg) and 10 μL Protein G Sepharose (Cytiva, #17061801) for 1 h. Bound proteins were analyzed by immunoblotting as described above. The recombinant proteins used in this study are listed in Supplementary Table 3.

**In vitro ubiquitination assay**. To prepare the glycosylated RNF13, HepG2 cells (6 cm dish) expressing HA-RNF13 were immunoprecipitated with anti-HA and eluted with 50 μL of 50 mM Tris-HCl buffer (pH 7.5). The immunoprecipitants (2 μL) were incubated with 100 nM E1 (UBE1; Merck, #SRP6147), 1 μM E2 (UBcH5c; Merck #662098), 5 μM ubiquitin (Merck, #U5507), 1 μM GAL9 (Wako, #074-06421), and 10 mM MgATP solution in 25 μL of E3 ligase buffer (R&D Systems, #B-71) for 1 h at 30 °C. The mixtures were subjected to CBB staining or immunoblotting with anti-UBE1 (Santa Cruz, #sc-53555) and anti-ubiquitin antibodies (Santa Cruz, #sc-8017). The recombinant proteins used in this study are listed in Supplementary Table 3.

**Immunofluorescence**. One day before transfection, HepG2 cells were seeded onto collagen-coated glass coverslips. At 48 h post-transfection, the cells were fixed with 4% paraformaldehyde, blocked with 10% normal goat serum (Thermo, #50062Z), and stained with primary antibodies[47]. Alexa Fluor-conjugated secondary antibodies were used to detect signals. The antibodies used in this study are listed in Supplementary Table 2. For staining of endomembranes, CellMask reagent (Thermo, #C37608) was added 1 h before fixation. Microscopic imaging was performed using an FV1000-D confocal laser scanning microscope (Olympus).

**Live cell imaging**. One day before transfection, HepG2 cells were seeded onto a collagen-coated glass-bottom dish (Matsunami, #D11134H). Subsequently, cells were transfected with vectors encoding GFP-LC3, pmCherry-GAL9, and HBc at a ratio of 1:1:1. Time-lapse images were acquired on Biostation IM-Q (Nikon) using a ×20 objective every 1 h for 48 h.

**Statistical analysis**. The statistical significance of differences between two groups was evaluated using a two-tailed unpaired $t$-test in Prism 8 software (GraphPad). Statistical significance was set at $P < 0.05$.

**Reporting summary**. Further information on research design is available in the Nature Research Reporting Summary linked to this article.

## Data availability
All other data are available from the authors upon request. Source data are provided with this paper.

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

## Acknowledgements
We thank Mina Dairaku, Kiho Tanaka, Kyohei Kurobe, Noriko Saido, Kyoko Ohnishi, and Kenji Yoshihara for their technical assistance. This work was supported in part by an AMED Grant-in-Aid for the Program on the Innovative Development and the Application of New Drugs for Hepatitis B (JP20fk0310103 to A.R. and JP20fk0310104 to K.M.).

## Author contributions
K.M. designed and performed the research, analyzed the data, and wrote the manuscript; M.N. and S.M. performed the research and analyzed the data; M.Ogawa., M.S., H.N., M.Ohnishi, K.W., and K.S. contributed reagents and analyzed the data; H.K. and T.W. analyzed the data; A.R. directed the research, analyzed the data, and wrote the manuscript.

## Competing interests
The authors declare no competing interests.
