## [Peer Review File · Nature Communications]

Galectin-9 restricts hepatitis B virus replication via p62/SQSTM1-mediated selective autophagy of viral core proteinsREVIEWER COMMENTS

Reviewer #1 (Remarks to the Author):

In this manuscript, Miyakawa et al identified an Interferon (IFN)-stimulated gene product, galectin-9 (GAL-9) as an HBV core protein (HBc)-interacting protein. The overexpression of GAL-9 induced the degradation of HBs through the p62/SQSTM1-mediated selective autophagy. Mechanistically, GAL-9 interacts with a ubiquitin-ligase RNF13 and drives autoubiquitination of RNF13 on the GAL-9-positive foci, leading to the recruitment of an adaptor protein for ubiquitinated proteins, p62/SQSTM1 and an autophagosomal localizing protein LC3 around the foci. Moreover, the authors further showed necessities of autoubiquitination activity of RNF13 and of the ubiquitin binding ability of p62 on the degradation of HBc. They further showed that knockdown of GAL-9 in primary human hepatocytes (PHHs) decreases an IFN-inducible antiviral effects. While this manuscript was straightforward and contained some interesting findings, the data presented in this manuscript were preliminary to prove the author's model.

Major comments

1. They should show both endomembrane structures and autophagosomes positive for GAL-9, RNF13 and p62 in HBc-expressing cells by immune-electron microscopic analysis.
2. In Figure 2E, signal for GFP-LC3 partially but not completely overlapped with GAL-9. Rather than immunofluorescent stain, correlative light electron microscopy is better to show autophagic degradation of HBc-GAL-9.
3. To prove the involvement of autophagy, the experiments with ATG-knock-down or knockout cells is indispensable. Only pharmacological experiments with BafA1 is insufficient.
4. Does RNF13 stably form in a complex with GAL-9? Alternatively, is RNF13 translocated onto GAL-9-positive structures after the interaction of HBc with GAL-9? The experiments with GAL-9 and RNF13 mutants, which are unable to interact with each other, would be useful.
5. Is the auto-ubiquitination of RNF13 induced upon the interaction with GAL-9? Do the authors have any evidence showing that p62 interacts with the auto-ubiquitinated RNF13?
6. In stress conditions including the impairment of proteostasis, the phase-separation of p62 is induced upon the interaction with ubiquitinated proteins, resulting in the increased number of p62-bodies (Sun D et al., Cell Res 2018). In the case of the p62-mediated HBc degradation, does the phase-separation for condensation of the cargo (i.e., HBc) occur? The authors should conduct the experiments to clarify this point (e.g., FRAP and Time-lapse etc).
7. p62 interacts with other ATG proteins such as FIP200 to generate isolation membrane around the cargos (Turco E et al., Mol Cell 2019). The authors should investigate the translocation of other ATG proteins around GAL-9-positive structures.

8. p62 LIR mutants (e.g., W341A L344A mutant), which are defective LC3-interaction are available to show the p62-mediated autophagic degradation of HBc.

9. In Figure 5, in addition of knock-down of GAL-9, the knock-down experiments for RNF13, p62 and ATG in PHHs are needed to prove the p62-mediated selective autophagy of HBc.

Minor comments

1. In a large number of experiments including immunoblot data, statistical analyses were missing. e.g., Figure 1B-D., Figure 2A, B, F and H., Figure 4G and Figure 5A.

2. In immunofluorescence analyses (Figure 2C, D, E, G, Figure 3D and Figure 4C), the authors should provide multiple cell images rather than single cell image.

3. In Figure 2C, HBc-single positive puncta were observed in cells expressing GAL-9. The authors should explain it.

4. In Figure 2G, the treatment of cells with IFN- β caused nuclear and cytoplasmic HBc-foci in a GAL-9 dependent fashion. What is the nuclear structures? Does GAL-9 localize to both nuclear and cytoplasm? In GAL-9 knock-down cells, nuclear LC3-puncta were formed upon exposure of IFN- β . What is nuclear LC3-positive structures?

Reviewer #2 (Remarks to the Author):

This manuscript address a fundamental aspect of HBV infection of cells and interferon-inducible genes (ISGs), which the authors found to include galectin-9, a carbohydrate-binding protein. The authors conclude that galectin-9 is involved in the process of xenophagy, a form of selective autophagy to protect against infection, by promoting degradation of the HBc, core protein of HBV. This autophagic mechanism appears to involve p62, a cargo receptor.

Galectin-9 was identified as a key player in this pathway by a BRET screen that it as a novel ISG that selectively triggers autophagic degradation of HBc.

However, as Gal-9 does not directly bind HBc, the authors found that Gal-9 binds P62, protein that can interact with ubiquitinated substrates, as well as RNF13, a ubiquitin ligase.

In many ways the results are interesting but relatively preliminary in many aspects. The authors propose that the Gal-9 induced by interferon as a result of HBV infection Gal-9 in some fashion may interact with

“...HBc-laden endomembranes” and that leads to its enhancement of HBc degradation. Moreover, the authors predict that intracellular Gal-9 has this function but this is not demonstrated directly. Although the authors state that “...GAL9 associated with at least two host proteins, p62 and RNF13..”, the data supporting this ‘association’ are not rigorous. Furthermore, the distinct mechanisms and interactions with autophagy receptors, as the authors imply, is also not established. The mechanisms of interactions of Gal-9 and indeed most galectins is through carbohydrate recognition, but this aspect of the biochemical nature of Gal-9 is not explored, simply hypothesized in the discussion.

Thus, the results indicate an indirect effect of the expression of Gal-9 in promoting HBc degradation through autophagic pathways. While the studies are indeed interesting, much more remains to be done to validate the essential elements of their conclusions.

What are the interacting molecules that are responsible for Gal-9 association in the BRET analyses? If Gal-9 does not “directly” bind either p62 and RNF13, which the authors overinterpret in their statement “...GAL9 interacts with RNF13”, to what does it bind?

The model on Fig. 6 is not strongly supported by the evidence at this point.

Does glycosylation of cellular material play any role, as proposed in this model?

Reviewer #3 (Remarks to the Author):

In this study, the authors aim to investigate the anti-HBV function of the IFN-inducible protein GAL9 and elucidate the underlying mechanisms. They claim that GAL9 inhibits HBV replication by indirectly recognizing HBV core protein HBc and mediating its clearance by p62-dependent selective autophagy. In addition, they propose that this process is regulated by autoubiquitylation of GAL9 interacting ubiquitin E3 ligase RNF13. While the identification of anti-HBV effect of GAL9 is potentially interesting, there are many defects in the experimental design. The mechanism analysis seems inadequate and the existing results and evidence are not strong enough to support their conclusions.

Major point

1. In most of the experiments including those for the key conclusions, the authors have just examined transfected HBc. To clarify the important points such as HBc degradation, GAL9-HBc interaction

(colocalization), and the role of RNF13 in HBc degradation, HBc in cells expressing HBV genome DNA and HepG2.215 cells should be checked.

2. The biggest drawback is that in the whole study they only observed the action and effect of overexpressed GAL9. In many of the key experiments, examinations on endogenous GAL9 are indispensable. HBc is diffusively distributed in the cytoplasm and only forms puncta under GAL9 overexpression or IFN β treatment suggesting that the punctate structures are formed by overexpressed GAL9. Then the story should not be "GAL9 detects HBc-containing endomembranes and activates...". Could the overexpressed GAL9 form puncta in cells without HBc expression?

3. Because GAL9 expression increased autophagosomes and p62 degradation (Fig. 2G and Fig. 3B), could GAL9 overexpression stimulates pan-autophagy instead of serving to detect HBc and mediating selectively HBc clearance? Under the same scenario, the data showing HBc-p62 colocalizations stimulated by GAL9 overexpression is very poor and endogenous p62 should be checked in the experiments, considering that overexpressed p62 itself can form puncta in cells.

4. What are the HBc-containing endomembrane compartments (HBc complexes)? GAL9 recognizes damaged membranes through binding membranous galactoside, what mediates HBc-GAL9 association? It seems like the authors are proposing that immunoprecipitation of HA-HBc pulls down the membrane structures, but they used a detergent-containing buffer for the immunoprecipitation (10 mM HEPES pH 7.4, 150 mM NaCl, 0.5% Triton-X-100).

5. In vitro RNF13 autoubiquitilation assay is required.

Minor points

1. Why are the HBc-puncta much less in GAL9 overexpression cells than in IFN β treated cells?

2. Fig. 2B, GFP-GAL9 should also be degraded with HBc by autophagy, then why its level in MG132-treated cells is much higher than in untreated and BafA1-treated cells?

3. Fig. 2 and Fig.5, the KD efficiency of GAL9 should be shown. In Fig. 2, the effect of GAL9 RNAi on HA-HBc level in IFN β treated cells is very weak.

Reviewer #4 (Remarks to the Author):

The study by Miyakawa et al. describes a new antiviral mechanism against hepatitis B virus (HBV), a chronic human pathogen of the liver affecting 350 million people worldwide. The authors demonstrate in elaborate experiments in virus-free assays that the HBV core protein HBc gets degraded through by autophagy involving the proteins galectin-9, RNF13 and p62. They start off their study using an elegant protein interaction assay (BRET) and discover galectin-9 as interaction partner of HBc. The majority of experiments are performed in a human hepatoma cell line and by ectopic expression of HBc and the respective host proteins. At the end of the study the authors show that some of their results, i.e. the

role of galectin-9 as antiviral factor, apply to primary human hepatocytes and in an infection setting. Finally, they confirm that galectin-9 and its antiviral activity are induced by interferon. Overall, the experiments are well designed and presented. For some experiments additional controls would be beneficial and for the imaging data quantitative analyses should be performed. The manuscript is very well written and results are clearly presented and discussed. Once the points below are addressed, I recommend acceptance of the manuscript for publication:

Major remarks:

1. Fig. 1. Very nice approach.
2. Fig. 2 C/G. Quantification of the observed co-localization is lacking and would strengthen the authors' conclusions.
3. Fig. 3A. A negative control using an isotype control antibody or using cells without Flag-Gal9 expression is missing. This additional negative control will be important to show the specificity of the IP. Ideally a vice versa IP would confirm the interaction of Gal9 with p62.
4. Fig. 3D. Quantification of co-localization is lacking.
5. Fig. 4B. A control IP from cells expressing RNF13, but lacking Flag-Gal9 is missing. Alternatively, isotype control IPs could be performed.
6. Fig. 4C. Quantification of co-localization is lacking.
7. Fig. 5I. Quantification of HBcAg and HBV cccDNA in the supernatant as in Fig. 5E/F would further strengthen the authors' conclusions of a Gal9 dependent reduction of HBV particle release.

Minor remarks:

1. Fig. 2C/D. It is not immediately clear why the authors choose to express a HT tagged protein as not BRET experiment is performed here.
2. Fig. 2F. The y-axis labeling is missing.
3. Fig. 5B. To exclude any effects of doxycycline treatment, a control with cells lacking tet-Gal9 and treated with doxycycline would be useful.
4. p.3, l. 68: "... replication of multiple types of viruses"
5. p.13, l. 295: "... exposed glycans on ruptured ..."
6. p.14, l. 299: "... Interaction of galectins with glycoproteins plays an important role ..."

Responses to the comments of Reviewer #1

In this manuscript, Miyakawa et al identified an Interferon (IFN)-stimulated gene product, galectin-9 (GAL-9) as an HBV core protein (HBc)-interacting protein. The overexpression of GAL-9 induced the degradation of HBs through the p62/SQSTM1-mediated selective autophagy. Mechanistically, GAL-9 interacts with a ubiquitin-ligase RNF13 and drives autoubiquitination of RNF13 on the GAL-9-positive foci, leading to the recruitment of an adaptor protein for ubiquitinated proteins, p62/SQSTM1 and an autophagosomal localizing protein LC3 around the foci. Moreover, the authors further showed necessities of autoubiquitination activity of RNF13 and of the ubiquitin binding ability of p62 on the degradation of HBc. They further showed that knockdown of GAL-9 in primary human hepatocytes (PHHs) decreases an IFN-inducible antiviral effects. While this manuscript was straightforward and contained some interesting findings, the data presented in this manuscript were preliminary to prove the author's model.

Response: We appreciate the insightful and constructive suggestions made by Reviewer #1. This reviewer noted that our manuscript was straightforward and contained some interesting findings, yet indicated that the data were preliminary. According to the reviewer's suggestion, we performed electron microscopy and studies with various mutants to strengthen our data and to prove the molecular mechanism of HBc degradation by GAL9-mediated autophagy. These findings have been included in the revised manuscript. Additionally, please review our point-by-point response to the reviewer's comments below.

#1-1. They should show both endomembrane structures and autophagosomes positive for GAL-9, RNF13 and p62 in HBc-expressing cells by immune-electron microscopic analysis.

Response: We appreciate this recommendation. Due to time and other constraints, we conducted immune-electromicroscopy for the most important factors in our manuscript, namely GAL9 and HBc. Our results clearly demonstrate the accumulation of HBc on GAL9-positive structures in cells (**Supplementary Fig. S3**). Additionally, we performed confocal laser microscopy showing that p62 and RNF13 were co-localized with GAL9 (**Fig. 4C, 5C**). Moreover, the membrane structure observed in the cell mask staining (Fig 2E) was similar to that of LC3 (**Supplementary Fig. S2D**), suggesting that this membrane component might be an autophagosome. Based on these observations, we have corrected our original working model. Please see our response below (#1-4).

#1-2. In Figure 2E, signal for GFP-LC3 partially but not completely overlapped with GAL-9. Rather than immunofluorescent stain, correlative light electron microscopy is better to show autophagic

degradation of Hbc-GAL-9.

Response: We thank the reviewer for this important suggestion. Although CLEM could not be performed on account of environmental constraints and technical barriers, we alternatively executed live cell imaging analysis in Hbc/GAL9-expressing cells to demonstrate the co-localization of GAL9 and LC3 over time. At most time points, including 8 h after transfection, GAL9 and LC3 were almost completely co-localized in the cells. These data have been added as **Supplementary Fig. S2E**.

#1-3. To prove the involvement of autophagy, the experiments with ATG-knock-down or knockout cells is indispensable. Only pharmacological experiments with BafA1 is insufficient.

Response: Based on this valuable suggestion, we confirmed that Hbc degradation by GAL9 did not occur in ATG5-depleted cells (**Fig. 2C**).

#1-4. Does RNF13 stably form in a complex with GAL-9? Alternatively, is RNF13 translocated onto GAL-9-positive structures after the interaction of Hbc with GAL-9? The experiments with GAL-9 and RNF13 mutants, which are unable to interact with each other, would be useful.

Response: We thank the reviewer for this important suggestion. GAL9-RNF13 seems to be the original complex partner, regardless of Hbc expression. In fact, in the absence of Hbc, RNF13 and GAL9 interacted with each other (**Fig. 5B**) and co-localized (**Fig. 5C**). Furthermore, we found that RNF13 was glycosylated intracellularly (**Supplementary Fig. S6C**), and the GAL9 association was completely dependent on RNF13 glycosylation (**Fig. 5D**). We also confirmed that RNF13 did not bind to the GAL9 mutant lacking the carbohydrate-recognition domain (CRD) (**Fig. 5E**). These data suggest that GAL9 recognizes the carbohydrate chains of RNF13 via the CRD. Based on our new findings, we would like to withdraw our original hypothesis that GAL9 recognizes damaged endo-membranes by binding to membranous galactoside. Our new schematic model was listed in new Figure 7.

#1-5. Is the auto-ubiquitination of RNF13 induced upon the interaction with GAL-9? Do the authors have any evidence showing that p62 interacts with the auto-ubiquitinated RNF13?

Response: The reviewer has raised an important point. Expression of a GAL9 mutant (Δ CRD), which cannot interact with RNF13, did not promote RNF13 ubiquitination (**Fig. 5H**), suggesting that the ubiquitination of RNF13 is induced upon the interaction with GAL9. Also, our in vitro

ubiquitination assay demonstrated that GAL9 indeed enhanced the auto-ubiquitination of RNF13 (**Fig. 5G**). These data suggest that RNF13 auto-ubiquitination is induced by its direct interaction with GAL9. In addition, auto-ubiquitination of RNF13 promoted the interaction of p62/SQSTM1 via its UBA domain for autophagosome formation (**Fig. 5H**). Taken together, our new findings indicate that RNF13 auto-ubiquitination is promoted by its interaction with GAL9, which leads to p62 recruitment.

#1-6. In stress conditions including the impairment of proteostasis, the phase-separation of p62 is induced upon the interaction with ubiquitinated proteins, resulting in the increased number of p62-bodies (Sun D et al., Cell Res 2018). In the case of the p62-mediated HBc degradation, does the phase-separation for condensation of the cargo (i.e., HBc) occur? The authors should conduct the experiments to clarify this point (e.g., FRAP and Time-lapse etc).

Response: Thank you for your valuable suggestion to examine the involvement of p62-mediated phase separation in HBc-foci formation. Indeed, phase separation of viral components is a novel conceptual framework in the mode of virus-host interaction. During our revision process, we identified viperin as a GAL9-binding factor involved in HBc-puncta formation (**Fig. 3**; for details, please see response #1-13). We further delineated the binding domain of viperin, in which the SAM domain and C-terminal region interact with GAL9 and HBc, respectively.

Our preliminary study demonstrated that a liquid-liquid phase separation (LLPS) marker G3BP1 was partially co-localized with GAL9-positive foci (Please see figure below). However, we believe that further investigations should be necessary to conclude this important question in our future study. We sincerely hope reviewer's very kind understanding on this point.

#1-7. p62 interacts with other ATG proteins such as FIP200 to generate isolation membrane around the cargos (Turco E et al., Mol Cell 2019). The authors should investigate the translocation of other ATG proteins around GAL-9-positive structures.

Response: As suggested, we performed an immunofluorescence analysis and confirmed FIP200 accumulation in the GAL9-positive structures of HBc-expressing cells (**Supplementary Fig. S5C**).

#1-8. *p62 LIR mutants (e.g., W341A L344A mutant), which are defective LC3-interaction, are available to show the p62-mediated autophagic degradation of HBc.*

Response: We appreciate this recommendation. Accordingly, we investigated whether the LIR-deficient p62 mutant could mediate HBc degradation by GAL9. Our data showed that re-expression of p62 LIR mutant in p62 knockdown cells could not rescue GAL9-mediated HBc degradation (**Fig. 4E**).

#1-9. *In Figure 5, in addition of knock-down of GAL-9, the knock-down experiments for RNF13, p62 and ATG in PHHs are needed to prove the p62-mediated selective autophagy of HBc.*

Response: As suggested, we performed HBV infection experiments using primary human hepatocytes (PHHs) with knockdown of RNF13, p62, or ATG5. Notably, our data demonstrated that IFN- β treatment was not able to suppress HBV production under any condition with knockdown of RNF13, p62, or ATG5. These data are now listed in **Supplementary Fig. S7D**.

Minor comments

#1-10. *In a large number of experiments including immunoblot data, statistical analyses were missing. e.g., Figure 1B-D., Figure 2A, B, F and H., Figure 4G and Figure 5A.*

Response: We apologize for this oversight. In the revised manuscript, we performed all appropriate statistical analyses and have added these data in each figure.

#1-11. *In immunofluorescence analyses (Figure 2C, D, E, G, Figure 3D and Figure 4C), the authors should provide multiple cell images rather than single cell image.*

Response: As requested, we have added multiple-cell images for each analysis. The figures have been updated accordingly in the revised manuscript (**Supplementary Fig. S2A, S2B, S2C, S2F, S4B, S4E, S4F, S5B, and S6B**).

#1-13. *In Figure 2C, HBc-single positive puncta were observed in cells expressing GAL-9. The authors*

should explain it.

Response: This is an important query. HBc single or multiple puncta were likely observed when only GAL9 was expressed or when treated with type I IFN. Since we observed HBc puncta more prominently in IFN- β -treated cells and GAL9 was unlikely to bind HBc directly (Supplementary Fig. S4A), we hypothesized the existence of another IFN-related factor(s) that facilitate HBc puncta formation. Using public databases, we selected IFN-related factor(s) that have been reported to bind GAL9 and examined whether they could also bind HBc. We found that an anti-viral factor, viperin could interact with both HBc and GAL9 (Fig. 3A-D) and regulate HBc-puncta formation by GAL9 expression (Fig. 3G, H). These results suggest that viperin might be a key mediator of the association between HBc and GAL9 for HBc-puncta formation. Please see new Figure 7 as our new schematic model.

#1-14. In Figure 2G, the treatment of cells with IFN- β caused nuclear and cytoplasmic HBc-foci in a GAL-9 dependent fashion. What is the nuclear structures? Does GAL-9 localize to both nuclear and cytoplasm? In GAL-9 knock-down cells, nuclear LC3-puncta were formed upon exposure of IFN- β . What is nuclear LC3-positive structures?

Response: We thank the reviewer for pointing this out. While HBc is known to be in the nucleus, LC3 and GAL9 are not. Therefore, the appearance of LC3 in the nucleus is likely noise due to the overlap of other cells. To avoid any confusion, we have replaced the picture with a more representative image (Fig. 2G).

Responses to the comments of Reviewer #2

This manuscript address a fundamental aspect of HBV infection of cells and interferon-inducible genes (ISGs), which the authors found to include galectin-9, a carbohydrate-binding protein. The authors conclude that galectin-9 is involved in the process of xenophagy, a form of selective autophagy to protect against infection, by promoting degradation of the HBc, core protein of HBV. This autophagic mechanism appears to involve p62, a cargo receptor. Galectin-9 was identified as a key player in this pathway by a BRET screen that it as a novel ISG that selectively triggers autophagic degradation of HBc. However, as Gal-9 does not directly bind HBc, the authors found that Gal-9 binds P62, protein that can interact with ubiquitinated substrates, as well as RNF13, a ubiquitin ligase.

In many ways the results are interesting but relatively preliminary in many aspects. The authors propose that the Gal-9 induced by interferon as a result of HBV infection Gal-9 in some fashion may interact with "...HBc-laden endomembranes" and that leads to its enhancement of HBc degradation. Moreover, the authors predict that intracellular Gal-9 has this function but this is not demonstrated

directly. Although the authors state that "...GAL9 associated with at least two host proteins, p62 and RNF13..", the data supporting this 'association' are not rigorous. Furthermore, the distinct mechanisms and interactions with autophagy receptors, as the authors imply, is also not established. The mechanisms of interactions of Gal-9 and indeed most galectins is through carbohydrate recognition, but this aspect of the biochemical nature of Gal-9 is not explored, simply hypothesized in the discussion. Thus, the results indicate an indirect effect of the expression of Gal-9 in promoting HBc degradation through autophagic pathways. While the studies are indeed interesting, much more remains to be done to validate the essential elements of their conclusions.

Response: We appreciate the helpful and constructive evaluations by Reviewer #2. Per your request for us to clarify the mechanism at the molecular level, we have focused on the molecular mechanism of GAL9-mediated HBc degradation in the revised manuscript. Specifically, we found that (1) viperin mediates GAL9 binding to HBc as an important cofactor, (2) GAL9 recognizes and binds glycosylated RNF13, thereby inducing its auto-ubiquitination, and (3) p62 is recruited to the ubiquitinated RNF13. Please review our point-by-point response to the reviewer's comments below.

#2-1. What are the interacting molecules that are responsible for Gal-9 association in the BRET analyses? If Gal-9 does not "directly" bind either p62 and RNF13, which the authors overinterpret in their statement "...GAL9 interacts with RNF13", to what does it bind?

Response: We appreciate this insightful comment. As pointed out by other reviewers (please see responses #1-13, #3-6), we observed more prominent HBc-puncta formation in IFN- β -treated cells than in cells expressing exogenous GAL9 (e.g., Fig. 2D and 2G). Together with our previous finding that GAL9 was unlikely to bind HBc directly (Supplementary Fig. S4A), this led us to hypothesize the existence of another host factor prior to its association with p62. Namely, there should be probably unidentified interferon-inducible proteins that supports the GAL9-HBc association and/or HBc-puncta formation. Using public databases, we screened IFN-related factor(s) that have been reported to bind GAL9, and further searched for those that also bind HBc. We identified viperin as a potent mediator of HBc and GAL9 (**Fig. 3A-D**). Viperin interacted with both HBc and GAL9 via distinct domains (**Fig. 3F**). In viperin knockdown cells, GAL9-HBc interaction was diminished and HBc-puncta formation by GAL9 was prominently inhibited (**Fig. 3G, H**). In addition, GAL9-mediated HBc degradation was suppressed in viperin-knockdown cells (**Supplementary Fig. S4D**). These results suggest that viperin might be involved in HBc-puncta formation through HBc and GAL9.

We also found that p62 binds to ubiquitinated RNF13, whose auto-ubiquitination is

promoted by GAL9 (for details, please see response #2-3). These results suggest that GAL9 and viperin are cooperatively involved in HBc degradation via poly-ubiquitination of RNF13, leading to p62-mediated autophagosome formation.

#2-2. *The model on Fig. 6 is not strongly supported by the evidence at this point.*

Response: Thank you for this insight. Based on our new findings, we have revised the schematic model for greater accuracy (**Fig 7**).

#2-3. *Does glycosylation of cellular material play any role, as proposed in this model?*

Response: Per your valuable suggestion, we have analyzed the role of glycosylation in our study. Previous reports indicated that galectin family proteins generally bind glycosylated proteins via carbohydrate recognition domains (CRDs). Indeed, we found that glycosylation of RNF13 is important for GAL9 binding to RNF13 (**Fig. 5D, E**). We also found that RNF13 is normally glycosylated in hepatocytes (**Supplementary Fig. S6C**). When RNF13 glycosylation was blocked, p62 did not associate with HBc/GAL9 puncta (**Fig. 5H**).

Responses to the comments of Reviewer #3

In this study, the authors aim to investigate the anti-HBV function of the IFN-inducible protein GAL9 and elucidate the underlying mechanisms. They claim that GAL9 inhibits HBV replication by indirectly recognizing HBV core protein HBc and mediating its clearance by p62-dependent selective autophagy. In addition, they propose that this process is regulated by autoubiquitylation of GAL9 interacting ubiquitin E3 ligase RNF13. While the identification of anti-HBV effect of GAL9 is potentially interesting, there are many defects in the experimental design. The mechanism analysis seems inadequate and the existing results and evidence are not strong enough to support their conclusions.

Response: We greatly appreciate the careful analysis and constructive suggestions made by Reviewer #3. The reviewer kindly suggested addressing a couple of important points to support our conclusions. We have revised the manuscript to address these concerns, as indicated below.

#3-1. *In most of the experiments including those for the key conclusions, the authors have just examined transfected HBc. To clarify the important points such as HBc degradation, GAL9-HBc interaction (colocalization), and the role of RNF13 in HBc degradation, HBc in cells expressing HBV*

genome DNA and HepG2.215 cells should be checked.

Response: We appreciate this suggestion. We conducted experiments using HepG2 cells transfected with the whole HBV genome as well as HepG2.215.7 cells stably expressing HBV genes. We confirmed GAL9-induced HBc degradation (**Supplementary Fig. S1B**), GAL9-HBc colocalization (**Supplementary Fig. S1C**), and GAL9-HBc interactions (**Supplementary Fig. S1D**) in these cells. We also confirmed that HBc degradation by GAL9 was abrogated by RNF13 knockdown (**Supplementary Fig. S6E**).

#3-2. The biggest drawback is that in the whole study they only observed the action and effect of overexpressed GAL9. In many of the key experiments, examinations on endogenous GAL9 are indispensable. HBc is diffusively distributed in the cytoplasm and only forms puncta under GAL9 overexpression or IFN β treatment suggesting that the punctate structures are formed by overexpressed GAL9. Then the story should not be “GAL9 detects HBc-containing endomembranes and activates...”. Could the overexpressed GAL9 form puncta in cells without HBc expression?

Response: We appreciate this helpful comment to elucidate the precise molecular mechanism of HBc-puncta formation. Based on the reviewer’s insightful suggestion, we attempted to analyze GAL9-induced HBc-puncta formation in more detail. Since endogenous GAL9 protein is hardly expressed in hepatocytes (see lane 1 of RT-PCR in Fig. 2A), we performed some additional experiments with IFN- β treatment in GAL9-knockdown cells (**Fig. 2G, 6G-I, Supplementary Fig. S7C**). Consistent with previous reports (Oncogenesis, 9, 65, 2020), GAL9 was localized primarily with punctate structures in the absence of HBc.

Further, as the reviewer pointed out in query #3-6, we found that HBc-puncta formation was more prominent when cells were treated with IFN- β as compared with GAL9 expression alone. Based on this observation together with the fact that GAL9 unlikely binds HBc directly, we hypothesized another regulator that mediate between GAL9 and HBc. We therefore screened GAL9 binding proteins and identified viperin as an HBc-interacting GAL9-binding protein that was responsible for HBc-puncta formation (**Fig. 3**; for details please see answer to query #3-6). Given these new findings, our original hypothesis that GAL9 detects and activates HBc-containing endocytic membranes has been withdrawn and proposed a new schematic (**New Figure 7**) model based on our findings in this revision.

#3-3. Because GAL9 expression increased autophagosomes and p62 degradation (Fig. 2G and Fig. 3B), could GAL9 overexpression stimulates pan-autophagy instead of serving to detect HBc and mediating selectively HBc clearance? Under the same scenario, the data showing HBc-p62

colocalizations stimulated by GAL9 overexpression is very poor and endogenous p62 should be checked in the experiments, considering that overexpressed p62 itself can form puncta in cells.

Response: We appreciate this comment. In the absence of HBc, GAL9-overexpression resulted in a slight decrease in endogenous p62 (**Supplementary Fig. S5A**), suggesting that GAL9 overexpression might induce pan-autophagy at a certain level. However, in the presence of HBc, GAL9 expression decreased endogenous p62 more prominently (**Supplementary Fig. S5A**). Furthermore, we confirmed the co-localization of HBc with endogenous p62 in GAL9-expressing cells (**Fig. 4C**). These data indicate the functional interaction between GAL9 and HBc may facilitate p62-mediated autophagosome formation.

#3-4. What are the HBc-containing endomembrane compartments (HBc complexes)? GAL9 recognizes damaged membranes through binding membranous galactoside, what mediates HBc-GAL9 association? It seems like the authors are proposing that immunoprecipitation of HA-HBc pulls down the membrane structures, but they used a detergent-containing buffer for the immunoprecipitation (10 mM HEPES pH 7.4, 150 mM NaCl, 0.5% Triton-X-100).

Response: We appreciate this insightful comment. In our revised paper, we have clarified the nature of the HBc-GAL9 interaction more clearly to directly address the inquiry raised by you and Reviewer #2 (#2-1). Indeed, our further screening revealed that viperin mediates the association of HBc with GAL9, thereby promoting RNF13 auto-ubiquitination and p62-mediated autophagy. These factors are cooperatively involved in HBc degradation. We also found that GAL9 recognizes and binds to the glycosylation site of RNF13 via the carbohydrate-recognition domain (CRD) (**Fig. 5D**).

Also, our immune-electromicroscopic analysis revealed that HBc was accumulated on GAL9-positive membrane structures in cells (**Supplementary Fig. S3**), suggesting that this membrane component might be an autophagosome. Based on these observations, we have corrected our original working model. Please see our response below (#1-1).

With these new findings, we have updated our schematic model, as shown in Figure 7.

#3-5. In vitro RNF13 autoubiquitination assay is required.

Response: As suggested, an RNF13 auto-ubiquitination assay was performed. Since GAL9 binds glycosylated RNF13, we utilized cell-associated RNF13 that was immunoprecipitated from the cell lysate. Our results indicate that GAL9 can enhance auto-ubiquitinated RNF13. These data are listed in the new **Fig 5G**.

Minor points

#3-6. *Why are the HBc-puncta much less in GAL9 overexpression cells than in IFNbeta treated cells?*

Response: The reviewer has raised an very important point. As described above, we assumed the existence of another IFN-related factor(s) involved in HBc puncta formation. We found that viperin interacted with both HBc and GAL9 and mediated between the molecules (**Fig. 3A-D**). In viperin knockdown cells, both GAL9-HBc interaction and HBc puncta formation by GAL9 were remarkably decreased (**Fig. 3G, H**). These results suggest that viperin may be involved in HBc-puncta formation by bridging HBc and GAL9. Notably, the expression of viperin and GAL9 was enhanced by IFN- β (**Supplementary Fig. S4C**), and it is likely that GAL9 and viperin cooperatively act to enhance the puncta formation and degradation of HBc in IFN- β -treated cells.

#3-7. *Fig. 2B, GFP-GAL9 should also be degraded with HBc by autophagy, then why its level in MG132-treated cells is much higher than in untreated and BafA1-treated cells?*

Response: It appears that MG132 treatment slightly upregulated GAL9 (**Fig 2B**), implying the involvement of the proteasome system in GAL9 turnover.

#3-8. *Fig. 2 and Fig.5, the KD efficiency of GAL9 should be shown. In Fig. 2, the effect of GAL9 RNAi on HA-HBc level in IFNbeta treated cells is very weak.*

Response: We agree with this point. The knockdown efficiency is now shown in **Fig. 2H** and **6H**. While GAL9 expression was slightly suppressed with siRNA, greater suppression occurred with shRNA due to its stable expression.

Responses to the comments of Reviewer #4

The study by Miyakawa et al. describes a new antiviral mechanism against hepatitis B virus (HBV), a chronic human pathogen of the liver affecting 350 million people worldwide. The authors demonstrate in elaborate experiments in virus-free assays that the HBV core protein HBc gets degraded through by authophagy involving the proteins galectin-9, RNF13 and p62. They start off their study using an elegant protein interaction assay (BRET) and discover galectin-9 as interaction partner of HBc. The majority of experiments are performed in a human hepatoma cell line and by ectopic expression of HBc and the respective host proteins. At the end of the study the authors show that some of their results, i.e. the role of galectin-9 as antiviral factor, apply to primary human hepatocytes and in an infection

setting. Finally, they confirm that galectin-9 and its antiviral activity are induced by interferon. Overall, the experiments are well designed and presented. For some experiments additional controls would be beneficial and for the imaging data quantitative analyses should be performed. The manuscript is very well written and results are clearly presented and discussed. Once the points below are addressed, I recommend acceptance of the manuscript for publication:

Response: We greatly appreciate the careful analysis made by Reviewer #4. The reviewer suggested that we address a couple of additional controls and/or statistical analysis to strengthen our manuscript. We have revised the manuscript to address these concerns, as indicated below.

Major remarks:

#4-1. Fig. 1. *Very nice approach.*

Response: Thank you for your kind comments.

#4-2. Fig. 2 C/G. *Quantification of the observed co-localization is lacking and would strengthen the authors' conclusions.*

Response: As requested, we have quantified co-localization in the revised manuscript (**Supplementary Fig. S2B, S2C, and S2F**). According to other reviewer's request (#1-11), another view of the cell image was added in Supplementary figures.

#4-3. Fig. 3A. *A negative control using an isotype control antibody or using cells without Flag-Gal9 expression is missing. This additional negative control will be important to show the specificity of the IP. Ideally a vice versa IP would confirm the interaction of Gal9 with p62.*

Response: We apologize for the lack of negative controls in this assay. As suggested, a negative control, using cell lysates without Flag-GAL9 expression, was added, and the updated figure has been included in the revised manuscript (**Fig 4A**).

#4-4. Fig. 3D. *Quantification of co-localization is lacking.*

Response: Per the reviewer's recommendation, we have now quantified co-localization in the revised manuscript (**Supplementary Fig. S5B**).

#4-5. Fig. 4B. *A control IP from cells expressing RNF13, but lacking Flag-Gal9 is missing.*

Alternatively, isotype control IPs could be performed.

Response: We have added a negative control using cells expressing RNF13 but without Flag-GAL9 expression (**Fig. 5B**).

#4-6. Fig. 4C. Quantification of co-localization is lacking.

Response: Per the reviewer's recommendation, we have now quantified co-localization in the revised manuscript (**Supplementary Fig. S6B**).

#4-7. Fig. 5I. Quantification of HBcAg and HBV cccDNA in the supernatant as in Fig. 5E/F would further strengthen the authors' conclusions of a Gal9 dependent reduction of HBV particle release.

Response: We appreciate this suggestion. HBV DNA in the supernatant and HBV cccDNA in the cells have been quantified in the revised manuscript (**Supplementary Fig. S7C**).

Minor remarks:

#4-8. Fig. 2C/D. It is not immediately clear why the authors choose to express a HT tagged protein as not BRET experiment is performed here.

Response: We appreciate this inquiry. To clarify, HaloTag-tagged proteins can be used in a wide variety of assays, including BRET experiments, Western blotting, and imaging analysis (ACS Chem. Biol., 3, 6, 373, 2008).

#4-9. Fig. 2F. The y-axis labeling is missing.

Response: We apologize for this oversight. Labels have now been added to the figure (**Supplementary Fig. S2C**).

#4-10. Fig. 5B. To exclude any effects of doxycycline treatment, a control with cells lacking tet-Gal9 and treated with doxycycline would be useful.

Response: As suggested, immunoblot analysis was performed with doxycycline-treated cells lacking tet-GAL9 (**Supplementary Fig. S7A**). The HBc protein level was not decreased by doxycycline treatment.

#4-11. p.3, l. 68: "... replication of multiple types of viruses ..."

Response: Thank you for pointing this out. This text has been corrected in the revised manuscript.

#4-12. p.13, l. 295: "... exposed glycans on ruptured ..."

#4-13. p.14, l. 299: "... Interaction of galectins with glycoproteins plays an important role ..."

Response: Thank you for pointing this out. These sentences have been removed for greater clarity because GAL9 seems to bind to the glycosylation site of RNF13, not glycans on the ruptured membrane (please see response #1-4).

REVIEWER COMMENTS

Reviewer #1 (Remarks to the Author):

The authors have addressed all my comments adequately and in my opinion the manuscript is suitable for publication in Nature Communications.

Reviewer #2 (Remarks to the Author):

The manuscript has been extensively revised with significant new data related to the mechanism by which it is proposed that galectin-9 through a xenophagic process, promotes degradation of the HBc, core protein of HBV, and this involves involve p62, a cargo receptor, and also two other key interacting molecules, the antiviral protein viperin and RNF13, a ubiquitin ligase. The evidence is strong the IFN induces Gal9 expression and that the effects through Gal9 are unique and not exhibited by other galectin family members, which are also expressed in the cytoplasm of cells.

This work has many novel insights that are important to the field. Yet, while the revisions are certainly important and improve the paper in terms of providing more insight into the mechanistic processes, the new data, which are extensive, raise a few questions. Some of the issues to address might not require more experimentation, but the authors should consider them in the revision.

1. For example, the evidence suggests that RNF13, a ubiquitin ligase, is N-glycosylated, based on the removal of glycans and molecular weight changes and changes in binding to Gal9. However, while the transmembrane RNF13 could be N-glycosylated on its luminal side, the cytoplasmic side, where it might bind Gal9 would not expect to have glycans. Thus, this key point needs to be resolved, as it hits directly at the mechanism. The results might imply that Gal9 is contacting RNF13 on the luminal side not in the cytoplasmic side.

2. Viperin is normally ER associated, but here the evidence is that HBc, gal9 and viperin form a cytoplasmic complex. Does expression of HBc deplete the ER of viperin, as well as consume most of the cytoplasmic Gal9?

3. The direct interaction of viperin with Gal9 is intriguing, but is it constitutive, or does it require the presence of HBc? In that regard is Gal9 constitutively associated with RNF13?

4. In terms of gal9 CRDs, it has two as the authors indicated, but they appear in other studies to have different carbohydrate recognition. Do the authors have insights into which of the CRDs of Gal9 are important? And does binding to RNF13 alter the oligomerization or stability of Gal9?

Reviewer #5 (Remarks to the Author):

As a fifth reviewer for the manuscript by Miyakawa et al, I did not perform a thorough review of the manuscript. Instead, I evaluated the authors' responses to reviewers #3 and #4 according to the editor's request.

Reviewer #4 indicated substantial enthusiasm for the original submission, but suggested that some experiments were performed with imperfect controls and that the confocal microscopy experiments suffered from a lack of quantitative analysis. For the most part, the authors were responsive to reviewer #4's critiques and I believe that this reviewer would likely support acceptance of the revised manuscript. One area where the authors could have been more responsive was to reviewer 4 point #3. For example, in Figure S1D, left panel, it is impossible to tell if Hbc is in protein complexes with Gal9 (as the authors propose) if the anti-Hbc antibody is just 'dirty'.

Reviewer #3 was much more critical of the original manuscript, and I do not believe that the authors adequately addressed this reviewer's primary criticism. Reviewer #3 stated that their main problem with the manuscript was an over-reliance on Gal9 over-expression. In response, the authors did show that the impacts of interferon treatment on several of the parameters that they tested were blunted by Gal9 knockdown. However, some of these data (e.g. Fig 2G) lacked quantitation and statistics. Also it seems as though these token efforts to show the role of endogenous Gal9 were somewhat insufficient.

Despite not doing a thorough evaluation of the authors' work, I do have a few of my own comments. First, I agree with reviewer 4 about the overall clarity of the manuscript, and I feel that the authors' Ifn-Gal9-Viperin-autophagy axis is novel and would be of interest to a wide audience. However, I have the following minor criticisms:

- 1) As presented, the in vitro ubiquitination assay (Fig 5G) is hard to interpret. The authors should provide blots showing the reaction components that were added or not and actually identify what antibodies are being used in the immunoblots.

2) In figure 3H, Hbc still forms puncta that show robust colocalization with over-expressed Gal9 following viperin knockdown. These data contradict the authors' model. Is this result because of inefficient viperin knockdown? Knockdown efficiency data are not shown.

Responses to the comments of Reviewer #1

“The authors have addressed all my comments adequately and in my opinion the manuscript is suitable for publication in Nature Communications.”

Response: We sincerely appreciate the helpful and constructive suggestions by Reviewer #1. We believe that our manuscript has been much improved by the changes made in accordance with these suggestions.

Responses to the comments of Reviewer #2

“The manuscript has been extensively revised with significant new data related to the mechanism by which it is proposed that galectin-9 through a xenophagic process, promotes degradation of the HBe, core protein of HBV, and this involves involve p62, a cargo receptor, and also two other key interacting molecules, the antiviral protein viperin and RNF13, a ubiquitin ligase. The evidence is strong the IFN induces Gal9 expression and that the effects through Gal9 are unique and not exhibited by other galectin family members, which are also expressed in the cytoplasm of cells.”

This work has many novel insights that are important to the field. Yet, while the revisions are certainly important and improve the paper in terms of providing more insight into the mechanistic processes, the new data, which are extensive, raise a few questions. Some of the issues to address might not require more experimentation, but the authors should consider them in the revision.”

Response: We wish to express our highest appreciation for the insightful suggestions made by Reviewer #2, which have helped us significantly improve the manuscript. We have addressed the comments of the reviewer as mentioned below.

(#2-1) For example, the evidence suggests that RNF13, a ubiquitin ligase, is N-glycosylated, based on the removal of glycans and molecular weight changes and changes in binding to Gal9. However, while the transmembrane RNF13 could be N-glycosylated on its luminal side, the cytoplasmic side, where it might bind Gal9 would not expect to have glycans. Thus, this key point needs to be resolved, as it hits directly at the mechanism. The results might imply that Gal9 is contacting RNF13 on the luminal side not in the cytoplasmic side.

Response: We acknowledge the valuable suggestion. We performed an immunofluorescence experiment using a membrane marker, CellMask, and found that partial co-localization of GAL9 and RNF13 may occur on the cytoplasmic membranes (**Supplementary Fig. S6c**), implying that the interaction may occur beside the membrane structure. However, it remains elusive whether the interaction occurs on the luminal or the cytoplasmic side.

Immunoelectron microscopy showed that HBe is located in the luminal areas whereas Gal9 mostly resides in luminal portions near the membrane (**Supplementary Fig. S3**). These observations suggest that GAL9 is contacting RNF13 on the luminal side rather than on the cytoplasmic side. On the

contrary, several previous studies have shown that another galectin family protein, GAL8, could associate with the glycans exposed on the luminal side upon membrane damage (Nature 482, 414, 2012; PLoS Pathogens e1006217, 2017). Therefore, GAL9 may also associate with glycosylated RNF13 via ruptured membrane. These implications have been addressed in Discussion (**Page 13**).

Additionally, while attempting to resolve this particular comment, we obtained deeper understanding of the mechanism and have improvised the schematic **Fig. 7A** with minor modifications. In the revised figure we show that HBc, Viperin and GAL9 form a complex which then associates with RNF13.

(#2-2) Viperin is normally ER associated, but here the evidence is that HBc, gal9 and viperin form a cytoplasmic complex. Does expression of HBc deplete the ER of viperin, as well as consume most of the cytoplasmic Gal9?

Response: We appreciate this insightful comment. We performed immunostaining with an endoplasmic reticulum (ER) marker. Consistent with a previous report (J Biol Chem. 297, 100824, 2021), viperin normally localized in ER (**Supplementary Fig. S4c**). However, interestingly, in cells expressing HBc, the staining for ER-localized viperin was reduced (**Supplementary Fig. S4c**), suggesting that HBc sequesters viperin from ER to another cytoplasmic membrane structure where it interacts with HBc and GAL9.

(#2-3) The direct interaction of viperin with Gal9 is intriguing, but is it constitutive, or does it require the presence of HBc? In that regard is Gal9 constitutively associated with RNF13?

Response: We thank the reviewer for this insightful suggestion. In fact, the finding that GAL9 binds both viperin and RNF13 has been reported in the literature and is now registered in a public database (BioGRID). However, in human hepatocytes as a natural target of HBV, GAL9 is rarely expressed except but is induced by type I IFNs (**Supplementary Fig. S4d**). We, therefore, propose a working model that the molecular interaction of GAL9 with viperin and RNF13 could be initiated upon host innate immune response or IFN treatment and, therefore, it may not be constitutive in human hepatocytes.

(#2-4) In terms of gal9 CRDs, it has two as the authors indicated, but they appear in other studies to have different carbohydrate recognition. Do the authors have insights into which of the CRDs of Gal9 are important? And does binding to RNF13 alter the oligomerization or stability of Gal9?

Response: We accept this as an important point. We examined the binding of GAL9 with RNF13 using deletion mutants lacking either N- or C-terminal CRD by immunoprecipitation. Our results showed that both GAL9 mutants were able to bind RNF13 (**Supplementary Fig. S6e**). However, GAL9 lacking both CRDs could not bind RNF13 (**Fig. 5e, Supplementary Fig. S6e**). These data suggest that either N- or C-terminal CRD indiscriminately associates with glycans on RNF13. In addition, the

results of native-PAGE showed that RNF13 apparently does not affect the oligomerization of GAL9 (Supplementary Fig. S6f).

Responses to the comments of Reviewer #5

“As a fifth reviewer for the manuscript by Miyakawa et al, I did not perform a thorough review of the manuscript. Instead, I evaluated the authors' responses to reviewers #3 and #4 according to the editor's request.

Reviewer #4 indicated substantial enthusiasm for the original submission, but suggested that some experiments were performed with imperfect controls and that the confocal microscopy experiments suffered from a lack of quantitative analysis. For the most part, the authors were responsive to reviewer #4's critiques and I believe that this reviewer would likely support acceptance of the revised manuscript. One area where the authors could have been more responsive was to reviewer 4 point #3. For example, in Figure S1D, left panel, it is impossible to tell if Hbc is in protein complexes with Gal9 (as the authors propose) if the anti-Hbc antibody is just 'dirty'.

Reviewer #3 was much more critical of the original manuscript, and I do not believe that the authors adequately addressed this reviewer's primary criticism. Reviewer #3 stated that their main problem with the manuscript was an over-reliance on Gal9 over-expression. In response, the authors did show that the impacts of interferon treatment on several of the parameters that they tested were blunted by Gal9 knockdown. However, some of these data (e.g. Fig 2G) lacked quantitation and statistics. Also it seems as though these token efforts to show the role of endogenous Gal9 were somewhat insufficient.”

Response: We sincerely appreciate the careful review of our manuscript by Reviewer #5 and are thankful for the constructive suggestions. The reviewer mentions that we need to address additional important points. Accordingly, we have revised the manuscript as indicated below.

1) As for the reviewer 4 point #3, we sincerely apologize for having missed the inclusion of a control in Fig. S1d. We performed immunoprecipitation with a non-immune IgG antibody as a control, and show that HBC forms a complex with GAL9 (Supplementary Fig. S1d).

2) To address the concern of reviewer 3, we performed additional key experiments to prove the role of endogenous GAL9. Because GAL9 is not expressed in hepatocytes in the steady state, we performed an experiment with specific siRNA following treatment with type I IFN. When endogenous GAL9 was suppressed, HBC did not show the co-localization with p62 even after IFN treatment (Supplementary Fig. S5c). This is another evidence that endogenously-induced GAL9 plays an important role in the selective autophagy targeting HBC.

As suggested, we have also added quantitative and statistical data in Fig. 2g.

“Despite not doing a thorough evaluation of the authors' work, I do have a few of my own comments. First, I agree with reviewer 4 about the overall clarity of the manuscript, and I feel that the authors'

Ifn-Gal9-Viperin-autophagy axis is novel and would be of interest to a wide audience. However, I have the following minor criticisms:"

Response: We again appreciate the careful evaluation by Reviewer #5. The reviewer mentions that our manuscript is interesting but requires additional solid data before publication. Our responses to the specific concerns are given below.

(#5-1) As presented, the in vitro ubiquitination assay (Fig 5G) is hard to interpret. The authors should provide blots showing the reaction components that were added or not and actually identify what antibodies are being used in the immunoblots.

Response: We apologize for the improper description in Fig. 5g. We have included additional images of CBB staining and immunoblotting for UBE1, UBCH5c, RNF13, and GAL9 (**New Fig. 5g**). The information about the antibodies has been clearly mentioned in the figure legend.

(#5-2) In figure 3H, Hbc still forms puncta that show robust colocalization with over-expressed Gal9 following viperin knockdown. These data contradict the authors' model. Is this result because of inefficient viperin knockdown? Knockdown efficiency data are not shown.

Response: We thank the reviewer for this insightful comment. In our previous study, the contradictory observation may be due to insufficient depletion of viperin by transient transfection of siRNA. Therefore, we performed additional experiments and the knockdown efficiency was increased up to 90% by determining the optimal concentration of viperin siRNA. This was proved by qRT-PCR analysis for viperin mRNA (**Supplementary Fig. S4h**). Immunostaining was then performed under these conditions and the co-localization of overexpressed GAL9 and HBc was prominently reduced compared with that in control (**New Fig. 3h**).

REVIEWERS' COMMENTS

Reviewer #2 (Remarks to the Author):

The manuscript has been extensively revised to adequately address all of the major concerns from the prior reviews. The results strongly support the discovery by the authors that galectin-9 (GAL9), which they show is IFN inducible, is involved directly in the autophagic degradation of the HBV core protein (HBc). This occurs through direct binding by GAL9 that also involves association with another antiviral factor viperin along with RNF13 and p62 in a carbohydrate-dependent manner requiring active CRDs of the GAL9. The novel anti-viral activity of GAL9 is extremely interesting and the pathway proposed raises many new questions for additional studies along this of the strong innate immune function of GAL9, and perhaps other galectins, as anti-viral agents.

Reviewer #5 (Remarks to the Author):

The authors have adequately addressed the concerns raised by reviewers #3 and #4 and by me.

Responses to the comments of Reviewer #2

“The manuscript has been extensively revised to adequately address all of the major concerns from the prior reviews. The results strongly support the discovery by the authors that galectin-9 (GAL9), which they show is IFN inducible, is involved directly in the autophagic degradation of the HBV core protein (HBc). This occurs through direct binding by GAL9 that also involves association with another antiviral factor viperin along with RNF13 and p62 in a carbohydrate-dependent manner requiring active CRDs of the GAL9. The novel anti-viral activity of GAL9 is extremely interesting and the pathway proposed raises many new questions for additional studies along this of the strong innate immune function of GAL9, and perhaps other galectins, as anti-viral agents.”

Response: We sincerely appreciate the helpful and constructive suggestions by Reviewer #2, which have helped us significantly improve the manuscript.

Responses to the comments of Reviewer #5

“The authors have adequately addressed the concerns raised by reviewers #3 and #4 and by me.”

Response: We wish to express our highest appreciation for the insightful suggestions made by Reviewer #5.